

# Hydrologic-Land Surface Modelling of a Complex System under Precipitation Uncertainty: A Case Study of the Saskatchewan River Basin, Canada

Fuad Yassin[1], Saman Razavi[1], Jefferson S. Wong[1], Alain Pietroniro[2], Howard Wheater[1]

[1]Global Institute for Water Security, University of Saskatchewan, National Hydrology Research Centre, 11 Innovation Boulevard, Saskatoon, SK, S7N 3H5, Canada
[2]National Hydrology Research Center, Environment Canada, 11 Innovation Boulevard, Saskatoon, SK, S7N 3H5, Canada

*Correspondence to*: Fuad Yassin (fuad.yassin@usask.ca)

**Abstract.** Hydrologic-Land Surface Models (H-LSMs) have been progressively developed to a stage where they represent the dominant hydrological processes for a variety of hydrological regimes and include a range of water management practices, and are increasingly used to simulate water storages and fluxes of large basins under changing environmental conditions across the globe. However, efforts for comprehensive evaluation of the utility of H-LSMs in large, regulated watersheds have been limited. In this study, we evaluated the capability of a Canadian H-LSM, called MESH, in the highly regulated Saskatchewan River Basin (SaskRB), Canada, under the constraint of significant precipitation uncertainty. The SaskRB is a complex system characterized by hydrologically-distinct regions that include the Rocky Mountains, Boreal Forest, and the Prairies. This basin is highly vulnerable to potential climate change and extreme events. A comprehensive analysis of the MESH model performance was carried out in two steps. First, the reliability of multiple precipitation products was evaluated against climate station observations and based on their performance in simulating streamflow across the basin when forcing the MESH model with a default parameterization. Second, a state-of-the-art multi-criteria calibration approach was applied, using various observational information including streamflow, storage and fluxes for calibration and validation. The first analysis shows that the quality of precipitation products had a direct and immediate impact on simulation performance for the basin headwaters but effects were dampened when going downstream. In particular, the Canadian Precipitation Analysis (CaPA) performed the best among the precipitation products in capturing timings and minimizing the magnitude of error against observation, despite a general underestimation of precipitation amount. The subsequent analyses show that the MESH model was able to capture observed responses of multiple fluxes and storage across the basin using a global multi-station calibration method. Despite poorer performance in some basins, the global parameterization generally achieved better model performance than a default model parameterization. Validation using storage anomaly and evapotranspiration generally showed strong correlation with observations, but revealed potential deficiencies in the simulation of storage anomaly over open water areas.

**Keywords:**

Precipitation Uncertainty, Hydrologic-Land Surface Models, multi-criteria calibration, storage and fluxes validation, Saskatchewan River Basin, Canada



# 1 Introduction

During the past few decades, Land Surface Models (LSMs) have expanded in scope and complexity. As they have become more sophisticated, they have increasingly integrated dominant hydrological processes, such as horizontal hydrological fluxes,

subsurface lateral water movement, and river flow routing, all of which are well recognized in the Hydrological Models (HMs) (Archfield et al., 2015; Davison et al., 2016). More recently, several LSMs have included irrigation and water management modules (Haddeland et al., 2006; Voisin et al. 2013a, 2013b; Pokhrel et al., 2017), which are well established in Global Hydrological Models (GHMs) (Döll et al., 2003; Archfield et al., 2015; Wada et al., 2017). The integration of these various processes has enabled LSMs to be used in support of a wide range of hydrological applications, in which they are referred to

as Hydrologic-Land Surface Models (H-LSMs) (Pietroniro et al., 2007). Although H-LSMs have made steady advances in representing hydrologic processes and incorporating human impacts on the terrestrial water cycle, the investigation of input data uncertainty and parameter estimation through calibration for large-scale basins has been limited and is not common practice with H-LSM models compared to their extensive use by the catchment hydrological modeling community. However, quantifying the sources of uncertainty and their magnitude associated with inputs and model simulations is essential to ensure

the effective use of the H-LSMs and increase the reliability of model predictions, especially for large-scale basins that are increasingly influenced by human activities. This paper addresses this important issue by evaluating the input and parameter uncertainty of an H-LSM with the use of multiple data sources and advanced calibration methods, using the Saskatchewan River Basin in western Canada, as a complex case study.

    H-LSMs require large numbers of parameters to describe vegetation, soil and snow processes. The parameterizations of H-

LSMs are often predefined by referring to look-up tables (Mendoza et al., 2015) and are rarely calibrated, particularly for large-scale basins and global scale modeling (Gupta et al., 1999; Davison et al., 2016). The limited application of parameter calibration is largely due to the difficulties of managing the large number of H-LSM parameters and the high computational requirements. Such difficulties escalate when a model is applied to a large-scale basin with complex surface heterogeneities and complicated hydrologic and water management features. Possible ways to deal with the challenges of computational

burden during calibrating H-LSMs are to reduce the number of parameters (Houser et al., 2001; Nasonova et al., 2009), use state-of-the-art computationally-efficient calibration methods (Tolson and Shoemaker 2007), and pre-emption strategies (Razavi et al., 2010), and surrogate modelling (Razavi et al., 2012).

    Recently, attention has been given to advance H-LSM parameter estimation through model calibration, but model calibration and fidelity assessment are commonly based on streamflow observations only and the models have been mainly applied to

smaller basins (Nasonova et al., 2009). Calibration only to streamflow can mask major model deficiencies, as different components, such as model structure, model parameters and forcing data can compensate for the shortcomings and error of each other, so that improvements to streamflow simulation calibration may be at the cost of degrading other model outputs. A





possible approach to address this issue is to include other sources of information available in addition to streamflow during calibration and validation (Crow et al., 2003; Lo et al., 2010; Yassin et al., 2017). For this purpose, remote sensing and local in situ observations can be used, such as the GRACE total storage anomaly (Tapley et al., 2004), MODIS-based evapotranspiration and land surface temperature (Mu et al., 2007; Zhang et al., 2010), and evapotranspiration estimates based

on flux tower latent heat flux observations (Barr et al., 2012).

Additional issues arise due to input data uncertainty, which again becomes more important for larger scale applications. Most H-LSMs use energy-based approaches that require high spatial-temporal resolution climate forcing data. The extent of forcing data required increases the significance of input uncertainties and their impact on the quality of H-LSM outputs. Although observational climate station data are often regarded as the "truth", they have limited functionality (applicability) in driving

H-LSMs for large basins. The density of meterological station observations is too sparse and the temporal resolution too course to sufficiently represent the spatiotemporal variability of climate forcing variables (Clark and Slater, 2006). Additionally, climate station measurements are prone to precipitation under-catch error, particularly for solid precipitation (Mekis and Hogg, 1999; Adama and Lettenmaier, 2003). The feasible option often chosen to drive H-LSMs for large-scale basins is to use gridded reanalysis climate forcing data derived from numerical weather prediction model output (Côté et al., 1998; Mesinger et al.,

2006; Sheffield et al., 2006; Weedon et al., 2014;) or use interpolated climate forcing based on climate station observations and radar data (Mahfouf, Brasnett, and Gagnon, 2007).

Although reanalysis and interpolated data provide better coverage of the space-time field of the meteorological conditions, they also contain significant errors that vary among products. The accuracy of each climate forcing product across different regions can have substantial impacts on the simulation of streamflow. For instance, Eum et al. (2014) showed that the runoff

from three high-resolution precipitation datasets was significantly different during the snowmelt period over high elevation alpine areas in the Athabasca River basin. Hence, the evaluation of the quality of available reanalysis climate forcing datasets over a given region is an important prerequisite not only to select the best performing forcing data but also to reduce error propagation in both H-LSM outputs and parameter estimation (Eum et al., 2014; Bajamgnigni et al., 2017).

The direct evaluation method, called "ground truthing", is to compare various reanalysis and interpolated products against

climate station observations for many points over a region of interest (Ebert et al., 2007; Wong et al., 2017). The indirect evaluation method uses different available climate forcing products separately to drive an H-LSM and then compares how well the various products simulate the streamflow (Eum et al., 2014; Bajamgnigni et al., 2017). The results of this indirect method, however, can be polluted by compensating effects, e.g., in parameterization. The indirect evaluation method is in general complementary to direct evaluation, because the streamflow at a gauge represents the integrated response of an upstream

watershed, and its comparison against simulated streamflows describes the integrated quality of the climate forcing product used. The indirect evaluation approach is more practical if applied using default model parameterization (without calibration), as then it will involve a reduced computational demand and also the results will not be complicated by the compensation effects of calibration.



The aim of this paper is to conduct a detailed analysis and evaluation of a physically-based H-LSM for a highly-managed, large-scale basin, using state-of-the-art calibration strategies and multiple data sources to enable quantification of modelling uncertainty. Such analysis is essential to comprehensively benchmark model performance, to examine water security vulnerabilities under future conditions, to serve as a test-bed (experimental basin) for the improvement testing of different

model process, and to evaluate new datasets. Additionally, such analysis helps to inform H-LSM applications for hydrologic operational forecasts and the management of large-scale basin water resources.

The specific objectives of this paper are as follows:

- To identify the most accurate precipitation dataset by evaluating error characteristics of multiple gridded precipitation datasets against ground-based observations.

- To evaluate the quality of gridded precipitation datasets in terms of how well they reproduce observations of multiple streamflow gauges when used to drive an H-LSM.

- To improve the H-LSM parameterization using a state-of-the-art computationally-efficient calibration approach, and evaluate the effectiveness of parameter transferability through validation in time and space, using independent multiple streamflow gauges not used in calibration.

- To test the model performance using multiple sources of observational information on model storage and output fluxes, and to ensure that the optimal parameters obtained are as realistic as possible (giving the "right answers for the right reasons") without error compensation across multiple outputs.

The study area of this paper is the 406,000 km$^2$ Saskatchewan River Basin (SaskRB) located in western Canada (Fig. 1). The SaskRB is currently the focus of an extensive research initiative due to its vulnerability to potential climate change and extreme

events such as droughts and floods (Razavi et al., 2015; Wheater and Gober, 2015). The basin presents a complex system characterized by hydrologically distinct regions that include the Rocky Mountains, Boreal Forest, and the Prairies, all of which affect the regional and global hydroclimate in unique ways. The distinct hydrology-land surface processes are further complicated by extensive water management (i.e., reservoir operation, diversion, and irrigation) and interjurisdictional water sharing policy (Islam and Gan, 2014; Wheater and Gober, 2015).

A limited number of hydrologic modeling studies have been conducted for different parts of the SaskRB. Wen et al. (2011) modelled agricultural drought using long-term soil moisture simulation over the Canadian Prairies using Variable Infiltration Capacity (VIC). VIC was calibrated using streamflow data and validated with observed soil moisture data. Tanzeeba and Gan (2011) used a Modified Interactions Soil-Biosphere-Atmosphere (MISBA) land surface scheme to study climate change impacts on some major sub-basins of the SaskRB. MISBA was calibrated using naturalized streamflow at sub-basin outlets.

Neither study accounted for reservoirs and irrigation, both of which significantly affect the simulation of flow and soil moisture. They also used only a limited number of streamflow stations for model calibration.

Recently, Faramarzi et al. (2016) applied the Soil and Water Assessment Tool (SWAT) model to assess model complexity and the effect of calibration strategy on freshwater estimation for Alberta watersheds, which includes the upstream sub-basins of the SaskRB. The authors compared the basin-outlet calibration to multiple-station model calibration, while model complexities



were changed by turning on and off some water management components. Their results showed that ignoring complex hydrological features resulted in inadequate model calibration and estimation of water resources. In their study, however, some of the cold region hydrological processes were represented in a simplified manner through conceptual models, such as the temperature-index approach for estimating snowmelt. Such simplistic process representation in cold regions may limit the

credibility of the resulting watershed model, in particular under a changing climate. A more detailed energy-based approach is preferable to allow estimation of individual components of the energy balance to simulate the energy fluxes within the snowpack. An energy-based approaches also allows direct estimation of evapotranspiration and sublimation and creates a stronger link to atmospheric modeling and remote sensing data (Overgaard et al., 2006). The present study therefore examines the modeling of the SaskRB using an energy-based and "process-based" H-LSM to calculate different hydrological

components using fine-time resolution climate forcing data along with physically-based model parameterizations.

For the reliable modeling of the current and future hydrology of the SaskRB, comprehensive model development and testing are needed that consider the representation of water management and distinct hydrological processes occurring in the basin. For this purpose, a Canadian H-LSM, MESH (Modélisation Environmentale communautaire - Surface Hydrology) was used. MESH has been used extensively for Canadian watershed research studies (Pietroniro et al., 2007; Haghnegahdar et al., 2014;

Haghnegahdar et al., 2017; Davison et al., 2016; Mengistu and Spence, 2016; Berry et al., 2017; Yassin et al., 2017) and recent developments have included reservoir operation (Yassin et al., 2019), irrigation, and flow diversion modules (the latter being incorporated for the first time in this study). This study is the first to report MESH model development for the entire SaskRB with representation of the aforementioned complexity and inclusion of detailed evaluation aimed to improve the understanding of the basin as a whole and create a test-bed for the simulation of alternative climate, land use and water management futures.

**2 Study area**

The Saskatchewan River Basin (SaskRB) (Fig. 1a) encompasses portions of the Canadian provinces of Alberta, Saskatchewan, and Manitoba, as well as a small portion of the US state of Montana. The SaskRB is situated in western Canada (98° - 118° W and 48° – 56° N), with a total drainage area of 406 000 km$^2$ and approximate maximum dimensions of 1300 km east-west and 700 km north-south. The source of the SaskRB originates in the eastern slopes of the Canadian Rockies in Alberta, which

includes parts of the Columbia Icefield. The two main tributaries of the SaskRB are the South and North Saskatchewan Rivers, both of which flow east and northeast through the Saskatchewan Prairies, before merging to form the Saskatchewan River, flowing through the Saskatchewan Delta (North America's largest freshwater inland delta), and draining into Lake Winnipeg in Manitoba. The two tributary river systems are further subdivided into nine subbasins of the Bow, Red Deer, Battle, Upper North, Central North, Lower North, Upper South, Lower South, and Eastern Saskatchewan rivers. The Upper South basin can

further be disaggregated into the Oldman basin and a small watershed draining from Montana, US.

The topographic elevation of the basin ranges between 218 and 3487 m above sea level (Fig. 1a). The physiographic characteristics extend from the rugged Canadian Rocky Mountains, foothills, and uplands on the far western side of the basin,





to lowlands and plains in the remaining parts of the basin. The Ecozones of the SaskRB (Fig. 1b) are classified into four ecozones; Montane Cordillera, Prairie, Boreal Plain, and Boreal Shield, covering 6, 58, 33, and 3 % of the basin area respectively. The Montane Cordillera Ecozone encompasses all the rugged mountains of the basin, the Boreal Plain has gently rolling to level topography, and the Boreal Shield contains hilly terrain with numerous ponds, wetland, and lakes. The Prairie

Ecozone covers post-glacial undulating plains to rolling plains and flat terrain with numerous depressional areas. The Prairie have several unique features: the pothole topography prevents some areas from draining to the major river system, the ecozone has internal drainage, and connection to the major river system are intermittent (Pomeroy et al., 2010, Shook et al., 2013). The parts of the ecozone not draining to the major river system are commonly called "non-contributing areas", defined as the drainage areas not playing a part in runoff in a flood that has a two-year return period (Godwin and Martin, 1975). Fig. 1c

shows the maximum possible ranges of the non-contributing areas in the SaskRB according to Prairie Farm Rehabilitation Administration (PFRA, Hydrology Division, 1983).

According to the Köppen climate classification, the SaskRB climate is classified as cold semi-arid and humid over the Prairies and cool and humid continental climate over the remaining parts, with a long, cold winter and a short, warm summer. The mean annual precipitation ranges from 601-800 mm in the Montane Cordillera to 201-600 mm in the Prairie and Boreal

Ecozones. Most of the precipitation comes as rainfall between April and August, and snowfall occurs in winter between October and April. Much of the annual runoff is generated from snowmelt during early spring from the Prairies and during summer from the mountains.

Figure 1b shows the land-cover of the SaskRB. The Montane Cordillera Ecozone is covered by barren land, glacier, grassland, shrubland, as well as needleleaf, broadleaf and mixed forest. Cropland and grassland dominate the land cover of the Prairies

followed by depressional wetlands and lakes. The Boreal Ecozone around the upland and eastern part of the basin is covered by needleleaf and broadleaf forest, while the central flat land is largely covered by cropland with sparse broadleaf forest. The soil type in the Prairies is dominated by Chernozemic soils, clay-rich soils with a high water-holding capacity favorable for agriculture. The Boreal Plain soil types are Brunisol and Luvisol, productive for crop and tree growth. Both the eastern and a portion of the northwestern parts of the basin are underlain by mineral soils.

The SaskRB is regulated by many reservoirs and diversions for hydropower production, irrigation, and other water supply. Irrigation is the major consumptive demand in the basin. In the Alberta portion, there are 13 irrigation districts providing water to 1 412 836 acres of farmland. In the Saskatchewan portion, 11 districts (80 000 acres) receive water from Lake Diefenbaker. Fig. 1d shows the major reservoirs and irrigation districts included in this study, and Tables S1 and S2 (supplementary materials), respectively, present brief summaries of the main reservoirs and irrigation districts.



# 3 Data

## 3.1 Model setup data

Topographic data were acquired from Canadian Digital Elevation (CDED, 2013) https://open.canada.ca/data/at a scale of 1:250,000. The land-cover map was extracted from the Canada Center for Remote Sensing (CCRS) (North American Land

Cover, 2005), which resulted in fourteen land-cover types for the SaskRB (Fig. 1b). The gridded soil texture data on percentage sand, clay, and organic matter from the Regridded Harmonized World Soil Database v. 1.2 (Wieder et al., 2014), were re-gridded and upscaled from the Harmonized World Soil Database (HWSD) v. 1.2 (Nachtergaele et al., 2009). Historical hydrometric data for model evaluation and calibration stations shown in Fig. 1c were obtained from the National Water Data Archive of the Water Survey of Canada. The list and brief details of the hydrometric stations are presented in the supplementary

materials Table S3. The irrigated cropland fraction was estimated using the Global Map of Irrigation Areas (GMIA) (Siebert et al., 2013). The data required for reservoir operation schemes such as maximum capacity, reservoir area-level-volume, time series of outflow-storage-inflow, and environmental release were obtained from the Water Survey of Canada, Saskatchewan Water Security Agency, and Alberta Environment and Parks.

## 3.2 Climate datasets

In Canada, ground measurements of precipitation are prone to different problems, among which gauge undercatch is of major concern (Mekis and Hogg, 1999). We used the Adjusted and Homogenized Canadian Climate Data (AHCCD) (Mekis and Hogg, 1999; Mekis and Vincent, 2011), and we considered this is the best available source of ground-based data for Canada. From AHCCD, a total of 15 stations located across the study area were chosen to provide evaluation benchmarks for five gridded precipitation products available for this study area (see Fig. 1c for location and Table S4 for climate station details).

A brief description of these five gridded precipitation products is provided in the following and Table 1, while more details are available in Wong et al. (2017).

The Canadian Precipitation Analysis (CaPA) was created to provide a data set of 6-hourly precipitation accumulations over North America from 2002 onwards at a spatial resolution of 15 km (Mahfouf et al., 2007). For Canada, the regional Global Environmental Multiscale (GEM) model was used to generate data by an optimum interpolation technique, in which the initial

guess from the model was updated by the rain-gauge measurements. CaPA has been continuously improving, assimilating data from the Canadian weather radar network and US radars near the border, and recently, refining its spatial resolution to 10 km (Fortin et al., 2015).

The Terrestrial Hydrology Research Group at the Princeton University had produced a global data set of 3-hourly meteorological data at a spatial resolution of 1.0° spatial resolution (~120 km) from 1948 to 2000 (Sheffield et al., 2006). The

components in generating this data set (called hereafter "Princeton") included the National Centers for Environmental Prediction-National Center for Atmospheric Research (NCEP-NCAR) reanalysis and a set of global observation-based data. Princeton has been updated and the 1901-2012 version at 0.5° and 3-hourly time steps was used in this study.



The European Union Water and Global Change (WATCH) Forcing Data methodology applied to the ERA-Interim (WFDEI) was developed to provide global data sets of sub-daily (3-hourly) and daily meteorological data at a spatial resolution of 0.5° (~50 km) covering the period of 1979 to 2012 (Weedon et al., 2014). Similar to Princeton, WFDEI was constructed based on the European Centre for Medium-Range Weather Forecasts Re-Analysis Interim product, combined with the Global

Precipitation Climatology Centre (GPCC) monthly variables and the Climatic Research Unit (CRU) monthly data. Therefore, WFDEI has used either GPCC or CRU precipitation totals to produce two sets of rainfall and snowfall data. In this study, both sets of data were used, for brevity, called hereafter GPCC and CRU.

The North American Regional Reanalysis product (NARR) was created to provide data sets of 3-hourly meteorological data for the North America domain at 32 km (~0.3°) spatial resolution from 1979 to 2015 (Mesinger et al., 2006). NARR was

derived from the NCEP-Department of Energy (NCEP-DOE) reanalysis and was combined with the NCEP regional Eta model, the Noah land-surface model, and numerous additional data sources. The assimilation of station observations over Canada has been discontinued since 2004 onwards.

To drive the MESH model and to evaluate precipitation uncertainty, the above precipitation data sets were individually combined with the other six required forcing variables (wind speed, air temperature, incoming shortwave radiation, specific

humidity incoming longwave radiation, and barometric pressure) gathered from the Global Environmental Multiscale (GEM) numerical weather prediction model (Côté et al., 1998). This is considered the best available source of these other forcing data, although it is recognized that there will be some inconsistencies with the precipitation fields from the other variables.

## 4 Description of MESH modelling system

### 4.1 MESH core components

MESH is Environment and Climate Change Canada's (ECCC)'s H-LSM framework (Pietroniro et al., 2007) encompassing several types of modeling structure (Fig. 2a). MESH uses an evenly space-distributed gridded spatial organization approach to configure the landscape. Sub-grid heterogeneity is represented by dividing each grid based on tiles defined by land-cover classes (in this study each land-cover types represent one tile) or based on other user-specified mosaic options. Each land-cover class has similar hydrological responses. MESH runs in off-line mode at a half-hourly time step using seven

meteorological forcing data variables. These are precipitation (mm s$^{-1}$), air temperature (K), wind speed (m s$^{-1}$), incoming shortwave radiation (W m$^{-2}$), incoming longwave radiation (W m$^{-2}$), specific humidity (kg kg$^{-1}$) and barometric pressure (Pa). MESH contains vertical, lateral (within grid) and grid-to-grid routing components. In the vertical, water and energy balances are calculated at tile scale with the Canadian Land Surface Scheme (CLASS) (Fig. 2b) (Verseghy, 1991, 2000; Verseghy et al., 1993). The basic prognostic variables of CLASS include the soil layers temperatures, the soil layers liquid and frozen

moisture contents; the mass, temperature, density, albedo and liquid water content of the snow pack; the temperature of the vegetation canopy and the mass of intercepted rain and snow present on it; the temperature and depth of ponded water on the soil surface; and an empirical vegetation growth index (Verseghy 2011). CLASS preserves the prognostic variables of each of



the tiles between time steps, while the surface fluxes are averaged using the tiles' fractional weight in each grid cell (Fig. 2c). CLASS contains four plant functional types, including needleleaf forest, broadleaf forest, crop, and grass; other vegetation units are approximated to one of the plant functional types by adjusting the parameter values. Glaciers are represented as a one-dimensional ice column. Water and energy balances are computed on the ice sheet using ice volumetric heat capacity and thermal conductivity. The default configuration of the CLASS soil layer contains three layers with a thickness of 0.10, 0.25, and 3.75m, respectively. The Green-Ampt method is used to estimate infiltration rate through the soil profile, and soil water storage and transmission to gravitational and soil moisture suction forces is calculated using a 1-D Richards' equation. The soil hydraulic properties are estimated using gridded soil texture data. The snowmelt process is governed by an energy budget approach. Evapotranspiration is estimated using a bulk mass transfer equation dependent on humidity (vapor pressure gradient). Interception of water and snow by vegetation is calculated as a function of plant leaf area index.

The lateral processes in MESH include: 1) blowing snow across tiles within a grid square, 2) lateral water movement in the soil, and 3) excess surface water flow from the tiles to the drainage system. Blowing snow transport and sublimation quantities are calculated within the grid square across tiles using the Prairie Blowing Snow Model (PBSM) (MacDonald et al., 2009). Wind-eroded snow from a tile either sublimates or transports and deposits into downwind tiles (according to aerodynamic roughness or drifting in a descending order) in the same grid square (MacDonald et al., 2009). Lateral movement of water in the soil and water on the surface is computed with either of the algorithms PDMROF (Mekonnen et al., 2014) or WATROF (Soulis et al., 2000). WATROF has been introduced to enhance the hillslope hydrology representation in MESH (Soulis, et al. 2000). Lateral flow in the soil is simulated as a function of lateral flow for unsaturated and saturated condition via the bulk value of the soil layer moisture by means of an approximated Richards' equation (Soulis et al. 2000, 2011). Surface overland flow is routed from the tile surface to drainage network within a model grid cell using Manning's approximation of the kinematic wave velocity equation. PDMROF has been introduced to calculate the variable contributing nature of Prairie regions. PDMROF is designed based on the probability density model (PDM) model of Moore (2007) to parsimoniously characterize the Prairies dynamic contributing area behaviour (Mekonnen et al., 2014).

The third lateral process, the routing component, includes grid-to-grid river flow routing and reservoir operation. It uses the WATFLOOD routing algorithm (Kouwen et al., 1993) to route the flows collected from overland flow, interflow and drainage through a watershed using a storage routing technique in which the outflow discharge is estimated as a function of water storage in the channel, computed using the continuity and Manning's equations. MESH reservoir operation along with irrigation and diversion is discussed in supplementary materials.

## 5 Modelling methodology

### 5.1 MESH model configuration

A grid resolution of 0.125° was used to configure MESH model, which resulted in 3667 grid cells for the SaskRB. Gridded watershed characteristics are derived based on the topographic data defined above (Fig. 1a). The flow direction and watershed





delineation was analysed at a terrain data native resolution of 1:50,000, and then the GreenKenue tool (Canadian Hydraulics Centre, 2010) was used to upscale to 0.125° modelling grids. The HWSD silt, sand, and clay percentage has a spatial resolution of 0.05°. The percentage of soil texture and land cover were regridded to 0.125° from their native resolution using area weights. Gridded irrigated cropland fraction was estimated using GMIA irrigation fraction and CCRS cropland fraction. The CCRS
cropland was separated into irrigated and non-irrigated cropland. The irrigation district boundaries (Fig. 1d) and model setup grids have been used as an intersecting layer while extracting the irrigated fraction GMIA. The data and parametrization of the DZTR reservoir operation scheme were obtained from Yassin et al. (2019), using storage, inflow, and release daily time series data for each reservoir. The non-contributing areas of the SaskRB (Fig. 1c) delineated by the Prairie Farm Rehabilitation Administration (PFRA, Hydrology Division, 1983) were used to configure grids as contributing and non-contributing.
PDMROF (Mekonnen et al., 2014) and WATROF (Solis et al., 2000) were used to drive the soil and surface water lateral movement in identified as the non-contributing areas and contributing areas, respectively.

**5.2 Evaluation of precipitation data uncertainty and model performance**

The precipitation and streamflow performance was evaluated using nine performance metrics that measure the goodness of fit on precipitation, different components of the hydrograph, evapotranspiration (ET), and total water storage anomaly (TWS).
The equations for the nine metrics are given in Table 3, where (1) $P_{bias}$ the percentage of precipitation bias between each precipitation data and station observation, (2) $P_{rmse}$ root mean square error to examine the magnitude of errors, (3) $P_r$ correlation coefficient, (4) $P_{\sigma G/\sigma R}$ standard deviation ratio to assess the amplitude of the variations, (5) $F_{bias}$ is the percentage of flow volume bias between simulated and observed flows, (6) $F_{nse}$ Nash Sutcliffe Efficiency on the streamflow, (7) $F_{lnse}$ is Nash Sutcliffe Efficiency on the logarithm of streamflow to put emphasis on fitting low flows, (8&9) $F_{rtws}$ and $F_{ret}$ are the correlation
coefficients for TWS anomaly and for evapotranspiration (ET), measuring the agreement between observed and simulated TWS and ET, respectively. The value of $F_{lnse}$ and $F_{nse}$ ranges between -∞ and 1.0 with an optimal value of one. The value of $P_{bias}$ and $F_{bias}$ varies between -100 % to +∞ with an optimal value of 0 %. A negative $P_{bias}$ and $F_{bias}$ indicate volume underestimation, and positive values indicate volume overestimation. The optimal value of $P_{rmse}$ is 0 mm day$^{-1}$ whereas that of $P_{\sigma G/\sigma R}$ is 1. A value of $P_{\sigma G/\sigma R}$ that is less than (greater than) one means the standard deviation of the precipitation data set is
smaller (larger) than that of the gauged precipitation data.

**5.3 Model-criterion optimization**

Model calibration was conducted using an aggregated multi-objective optimization approach. The aggregation uses equal weights to combine three objective functions defined on streamflow $F_{bias}$, $F_{nse}$ and $F_{lnse}$, as shown in Eq. (1). To attain a spatially-consistent model performance, each objective function is evaluated on multiple calibration streamflow stations
individually, and then averaged to provide a basin-wide performance for use in calibration; this approach is commonly referred

to as the "global calibration" approach. Global calibration is regarded as a reasonable trade-off between local performance and regional consistency of parametric information (Haghnegahdar et al., 2014; Ricard et al., 2013).

$$\underset{x\in\theta}{\text{minimize}}\, F(x) = \underset{x\in\theta}{\min}\left\{\left[\frac{\sum_i^m abs(F_{bias}(x)_i)}{m}\right] + \left[\frac{\sum_i^m -1*F_{nse}(x)_i}{m}\right] + \left[\frac{\sum_i^m -1*F_{lnse}(x)_i}{m}\right]\right\} \qquad (1)$$

$\theta$: Parameter space, $x$: model parameters (decision variables), $m$: total number of streamflow stations averaged

The number of MESH parameters that require calibration in MESH increases with the number of land-cover classes, complicating the parameter identification and equifinality challenges. Calibrating a large number of soil, vegetation and snow

parameters only based on streamflows can lead to significant errors in other model state variables and other flux outputs (Yassin et al., 2017). Improved calibration can be achieved by using multiobjective-multivariate calibration with a large number of model evaluations. Running MESH for the SaskRB for seven years in series requires ~10 hours, and to complete 5 000 model evaluations on a parallel computing system with 10 nodes with 16 cores per node needs around two weeks of computational time. Such a high computational demand creates difficulty to conduct multivariate calibration that requires more

than 20 000 model evaluations.

Alternatively, this study calibrated a sub-set of parameters chosen using the previous sensitivity analysis results on MESH (Yassin et al., 2017; Haghnegahdar et al., 2017) (Table 2). The selected sub-set includes parameters of river routing, some of the CLASS vegetation parameters, and the conceptual parameters of PDMROF and WATROF. These parameters were shown to control most of the variability in the model performance (Haghnegahdar et al., 2017; Yassin et al., 2017;) and are shortlisted

here as essential parameters for calibration, while other 'less influential' parameters were set at their default values. This reduction in the calibration problem helped us reduce the associated computational burden, while obtaining consistent and well-performing values for the parameters that dominantly control the model behavior. The results from a model run using default parameters (a priori) are used to benchmark model performance improvement through calibration. The parameter values and bounds (Table 2) in both configurations were specified based on suggested values from the CLASS technical

manual (Verseghy 2011), previous studies and other H-LSMs suggested values.

The calibration of the selected model parameters was conducted using the Dynamically Dimensioned Search (DDS) method (Tolson and Shoemaker 2007). The DDS algorithm has been demonstrated to have advantages over other optimization approaches when the model is computational intensive, because it requires a relatively low number of model evaluations (<10 000) to achieve a good global solution (Tolson and Shoemaker 2007). This is a very important factor and hence the algorithm

is well suited for the MESH model, which is computationally intensive; applying an ideal number of model evaluations for convergence of the optimal solution or Pareto front for MESH would be extremely time consuming particularly when the model is run on a large watershed (e.g. the SaskRB) with a large number of model parameters (decision variables).

### 5.4 Model calibration and validation configuration

Model calibration and validation were carried out using data from a 10-year period. An initialization test was conducted to

determine the required warm-up period, and the results showed (not reported here) that a one year warm-up brings the


prognostic variables close to stability. The first year (2002) was used for model warm-up, the next six years (2003-2008) for model calibration and the last three years (2009-2011) for model validation. Thirty-seven streamflow stations available through the Water Survey of Canada for SaskRB have been used for calibration and validation purposes (Fig. 1d and Table S3). These stations were screened based on two criteria, (1) having drainage areas above 1500 km$^2$, and (2) streamflow data record lengths

at least for the model calibration period (2003-2008). Twenty-two out of thirty-seven stations (60 %) have been used for model temporal calibration and validation, while the remaining 15 (40 %) stations were used for spatial validation (independent stations for validation). Drainage areas of the stations used for calibration ranged from approximately 1500 to 289 000 km$^2$, and of those used for spatial validation ranged from 2580 to 389 000 km$^2$.

As an additional way of model validation, the simulated total water storage anomaly (TWS) and evapotranspiration (ET) were

evaluated against corresponding available observations. The modelled TWS were compared against TWS anomaly observation by GRACE satellite as validation. For this study, the GRACE TWS anomaly data were obtained from Natural Resources Canada in which the GRACE TWS data is estimated by means of a two-step filtering approach (Huang et al., 2012; Lambert et al., 2013) at 1° × 1° resolution for the period of 2003-2011. The TWS anomaly of the GRACE data is relative to the mean of January 2003 to December 2009.

Furthermore, we evaluated the simulated evapotranspiration using two observed data sets: (1) global NDVI-based gridded monthly evapotranspiration data (Zhang et al., 2010) in which NDVI is used with surface resistance to estimate transpiration and soil evaporation using a modified Penman-Monteith method and open water evaporation using Priestley-Taylor approach. These data are available from 1983 and more details about the estimation algorithm are provided in Zhang et al. (2010), (2) ET estimated based on latent heat flux observation for two flux towers, namely, "Old Jack Pine (OJP)" and "Old Black Spruce

(OBS)" located in the Boreal plain of the SaskRB near the basin outlet. The flux tower observations are available for the period of 1999-2009.

## 6 Results and Discussion

The results are presented here in four sections. Sect. 6.1 shows comparisons of the precipitation data with adjusted precipitation gauge measurements. Sect. 6.2 discusses the performance of each precipitation data set for seasonal streamflow simulation.

Sect. 6.3 presents the model calibration and validation results. Lastly, sect. 6.4 displays the performance of the calibrated model for other fluxes and stores as additional validation.

### 6.1 Direct precipitation data inter-comparison

The abilities of gridded precipitation products to represent daily variability of precipitation amounts were examined using four performance measures: percentage of bias ($P_{bias}$), root mean square error ($P_{rmse}$), correlation coefficient ($P_r$), and standard

deviation ratio ($P_{\sigma G/\sigma R}$), as shown by Eqs (1) to (4) in Table 3. Four aspects of the gridded precipitation products were assessed,





with accuracy of product estimation by $P_{bias}$, magnitude of the errors by $P_{rmse}$, direction and strength of the linear relationship between gridded products and precipitation-gauge station data by by $P_r$, and amplitude of the variations by $P_{\sigma G/\sigma R}$.

When examining the accuracy (Fig. 3a), in the Montane Cordillera Ecozone where the Rocky Mountains lie (stations I, IV, and VI), Princeton and CaPA were associated with a generally negative $P_{bias}$ in four seasons. In particular, Princeton and CaPA

greatly underestimated the precipitation amounts in the Upper South Saskatchewan River basin in spring by >-23.9 % and >-32.6 % and in winter by >-10.0 % and >-12.4 %, respectively. CRU and GPCC had similar performances in the mountain stations where they performed particularly well in summer and autumn in the Bow River basin. NARR, on the other hand, overestimated the precipitation amounts in the Bow River basin in spring and winter by 49.2 % and 93.6 %, respectively. Along the boundary between the Montane Cordillera and Prairie Ecozones (stations II and VII), Princeton and CaPA performed

the best in autumn in which the accuracies were within 0.4 to 2.4 %. CRU and GPCC overestimated the precipitation amount in autumn (>11.7 % and >15.6 %), and NARR consistently showed a positive $P_{bias}$ for the individual seasons (from spring to winter: >51.0, >11.6, >34.2, and >28.3 %). Within the Prairie Ecozone (stations IX and X), all data sets showed negative $P_{bias}$ in summer with a range of -3.0 (CaPA) to -34.7 % (Princeton). In the Central North Saskatchewan River basin where the Prairie and Boreal Plain Ecozones meet (stations 3, 5, and 8), all datasets generally showed positive $P_{bias}$ in summer (except

CaPA) and negative $P_{bias}$ in spring and winter (except NARR). Within the Boreal Plain Ecozone (station XI), all gridded datasets mostly underestimated the precipitation amounts for the individual seasons by >-9.0, >-0.6, >-1.3, and >-8.0 %, respectively. Moving downstream towards the Lower Saskatchewan River basin, all products had similar performance in all seasons in which the precipitation amounts were overestimated in the Boreal Shield Ecozone (station XIII) and underestimated in the Boreal Plain Ecozone (station XV). Overall, CaPA showed negative $P_{bias}$ in which the precipitation amounts for the

individual seasons were underestimated by >-11.4, >-16.6, >-2.7, and >-0.3 % respectively (Table 4). CRU and GPCC had similar performance in most of the stations. While NARR generally showed the overall largest overestimation of precipitation amounts in all stations (5.7 %), Princeton provided the overall largest underestimation (-15.3 %).

In terms of the magnitude of errors (Fig. 3b), the performance of all datasets generally was the worst in the mountain stations (especially in the Upper South Saskatchewan River basin), in which the overall $P_{rmse}$ was the highest for the individual seasons

(from spring to winter: 5.44, 6.66, 4.49, and 2.98 mm day$^{-1}$). In addition, all datasets performed particularly poorly in summer in the Red Deer River basin (station VII), where the magnitude of errors ranged from 5.67 to 9.20 mm day$^{-1}$. In general, Princeton showed the worst correspondence with gauge precipitation data, providing the overall highest $P_{rmse}$ over all stations in the individual seasons (from spring to winter: >5.22, >8.25, >5.47, and >2.68 mm day$^{-1}$). Moreover, referring to Table 4, Princeton had the lowest $P_r$ in all stations in four seasons (0.13, 0.17, 0.18, and 0.11). Conversely, the performance of CaPA

was the best in both correlation with observations and magnitudes of errors regardless of station or season, with the grand $P_r$ and $P_{rmse}$ of 0.73 and 3.09 mm day$^{-1}$, respectively. CRU, GPCC, and NARR shared similar $P_r$ and $P_{rmse}$ for all stations and seasons, and their performance were in between Princeton and CaPA.

Regarding the amplitude of variations (Fig. 3d), the variations of all datasets generally were much smaller than those of gauged precipitation data during winter in the Upper South Saskatchewan River Basin (station IV), in which $P_{\sigma G/\sigma R}$ ranged from 0.29





(CRU) to 0.54 (NARR). All datasets also had too little variability in spring in the Boreal Plain Ecozone (station XI) where $P_{\sigma G/\sigma R}$ varied from 0.51(NARR) to 0.74 (Princeton). In particular, Princeton estimated $P_{\sigma G/\sigma R}$ the best in the Central North Saskatchewan River Basin (stations III and V) in spring (0.90 – 1.00) but had much larger variability in summer (1.60 – 1.73) and autumn (1.41 – 1.88). Princeton also estimated $P_{\sigma G/\sigma R}$ the best in spring in several Ecozones, the Prairie (stations IX and

X), the Boreal Plain (station XI), the Boreal Shield (station XIII), and the Boreal Plain (station XV). CRU and CaPA consistently had too little variability at all stations (especially in the Upper South Saskatchewan River basin) in spring and winter, while NARR showed even much smaller $P_{\sigma G/\sigma R}$ in the Boreal Plain Ecozone (stations XI and XII) and the Lower Saskatchewan River basin (stations XIII to XV) in the four seasons. The standard deviations of GPCC were most similar to those of the precipitation-gauge station data in autumn in these zones: the Boreal Plain Ecozone (stations XI and XV), the

Boreal Shield Ecozone (station XIII), and the transition zones between the Montane Cordillera and Prairie Ecozones (station II) and between the Prairie and Boreal Plain Ecozones (station V).

The preceding discussion evaluated the relative performance of five precipitation data sets over the SaskRB relative to gauge measurements in four seasons. Based on the assessment, it is not surprising that all products failed to accurately represent the precipitation amounts and variabilities at high altitude where complex terrain is found (i.e., the Rocky Mountains). The

performance was particularly poor in spring when the precipitation phase changes and in winter when solid precipitation occurs (Schirmer and Jamieson, 2015). Furthermore, in regions where two Ecozones transition, such as between the Montane Cordillera and Prairie Ecozones and between the Prairie and Boreal Plain Ecozones, systematic underestimation (Princeton and CaPA) or overestimation (NARR) in four seasons was witnessed. Our analysis echoed previous studies (Mesinger et al. 2006; Wong et al. 2017) that showed that NARR did not perform well in the Saskatchewan River basin, perhaps due to not

assimilating precipitation gauge measurements over Canada since 2004. Despite its general underestimation of precipitation amounts across the basin, CaPA performed well by showing a minimum magnitude of precipitation errors and capturing the timing. The reason for the overall underestimation could be that the version used in this study (2.4β8 version) did not consider the elevation of the stations when interpolating precipitation data and did not assimilate radar network data (Fortin et al. 2018). However, the quality of CaPA in complex terrain and in winter is expected to improve because in the latest version of the

model radar data has been assimilated (Fortin et al. 2015) and upcoming developments to the product have been identified (Fortin et al. 2018).

Because of the limitations of the adjusted precipitation-gauge stations, the evaluations of different precipitation data sets in this study should be interpreted with care (Wong et al. 2017). Since many of the stations in the SaskRB lacked complete temporal coverage of the entire study period, only 15 stations from the AHCCD dataset could be used for evaluations. Also,

the stations are sparse across the basin such that only one or two are found within each subbasin (or each Ecozone). In addition, the data of these 15 stations were already used in developing some of the precipitation datasets (e.g., CaPA). Thus, this study assessed the fit to observations of the datasets instead of their spatial interpolative skill. Nonetheless, given the difficulties in obtaining independent observations and the assumption that the AHCCD serve as the best estimates of actual precipitation,



evaluation of these data, as well as comparison of different precipitation products, are still valuable in providing the error characteristics of the dataset across the basin and offering useful information for subsequent hydrological modelling.

**6.2 Precipitation data intercomparison based on streamflow simulations**

The streamflow performance metrics for different precipitation data were compared by keeping model configuration and parameterizations the same; the only change factor was the precipitation data. The comparison based on streamflow complements the previous section as streamflow represents the integrated response of an upstream watershed to describe the integrated effect of precipitation data quality. The streamflow simulation comparison was evaluated without calibration using default (a priori) parameter values for two main reasons: 1) To minimize the effect of mixing with and compensating for other errors arising from the process representation, model structural uncertainties, and parameter uncertainties; and 2) to evaluate the performance of different precipitation data sets on streamflow simulations before calibrating the model with all precipitation products, which is very computational demanding. Evaluating the data sets' first enabled us to select the best performing precipitation data for subsequent calibration.

Regarding the predictive power of the model (Fig. 4a), in the Upper North Saskatchewan River basin, the poor performance we observed before at the mountain precipitation-gauge stations (stations I and IV in Fig. 3b) had an immediate impact on the model performance such that the $F_{nse}$ of the headwater stream gauges (stations 1, 2, and 5) were the lowest (0.04, 0.19, and 0.18) across the basin. The model performance gradually improved when going downstream with $F_{nse}$ values increased from 0.39 (station 3) to 0.45 (station 6). In particular, GPCC and CaPA performed better than the other precipitation data sets in spring and summer. The model performed similarly in the Bow River basin where the $F_{nse}$ values of the headwater streamflow gauges (stations 9 to 12) were below 0.4, and those of the downstream gauges (stations 13 to 16) were above 0.6. The poor performance in the Lower South Saskatchewan River basin ($F_{nse}$ values all below zero in all seasons except station 35 in autumn) was also probably due to the direct impact of the high magnitude of error in the precipitation products (station IX in Fig. 3b). Conversely, despite the high $P_{rmse}$, especially in summer (station VII in Fig. 3b), the model could still perform well for all stations, particularly in the upper part of the Red Deer River basin's stations 17 and 19 (both $F_{nse}$ greater than 0.5). For the other subbasins in the SaskRB, GPCC and CaPA generally generated the best streamflow performance in all seasons except winter. CaPA had the lowest $P_{rmse}$ and resulted in the best model performance, with the grand $F_{nse}$ of 0.35. However, Princeton had the overall lowest $F_{nse}$ in most stations and seasons, with the grand $F_{nse}$ (-0.81). The precipitation comparison based on $F_{lnse}$ showed a similar pattern of variability but generally inferior to that of $F_{nse}$. For the sake brevity, the results of $F_{lnse}$ are not further discussed.

As for the accuracy of the model performance (Fig. 4b), all precipitation products (except NARR) consistently generated a negative $F_{bias}$ for the Battle River basin (stations 29-32) in all seasons. This result suggests that in this basin precipitation errors play a dominant role in affecting the model accuracy in streamflow simulation. Although there are no AHCCD precipitation gauges in the Battle River basin and therefore no direct assessment could be done, the above precipitation analysis showed that NARR generally overestimated the precipitation amounts while other data sets provided different degrees of





underestimation. The degree of accuracy of the streamflow simulation reflected that of the precipitation products. A similar positive association between the accuracies of precipitation products and their associated model performances was witnessed in the headwater of the Red Deer River basin where NARR generally showed a positive $F_{bias}$ (station 18) for the individual seasons (overall $F_{bias}$ of 21%), and a similar $P_{bias}$ (station VII in Fig. 3a).

Likewise, the general underestimation of precipitation amounts in Princeton and CaPA in four seasons directly propagated and affected the streamflow simulation in the headwater of the Upper South Saskatchewan River basin in which every $F_{bias}$ was negative (-67 % and -35 % overall average underestimation for Princeton and CaPA, respectively). In particular, the general overestimation and underestimation of streamflow simulation at station 5 followed the overestimation and underestimation of precipitation amounts by different precipitation products at station IV (Fig. 3a) for the individual seasons. However, such

seasonally positive association was not strong in other parts of the headwaters, where the overestimation of precipitation amounts by CRU and GPCC in winter (station I in Fig. 3a) was followed by the underestimation of streamflow simulation in winter and overestimation in spring (stations 2 and 3). This could possibly be due to the lag time of the hydrological response to precipitation forcing. Furthermore, this positive association was dampened when going downstream and did not hold in other subbasins.

Most of the streamflow simulation was overestimated in the Upper North Saskatchewan River basin (station 26) in spring and winter, despite the underestimation of precipitation amounts in the data sets (stations III and V in Fig. 3a). This overestimation is similar to that of the Lower North Saskatchewan River basin. This contradiction could be due to (1) the error propagated from the mountains in the Upper North Saskatchewan where a large portion of the flow is generated, and (2) the mixing effects of error from the precipitation products and other model errors. Additionally, the streamflow simulation was consistently

underestimated in all seasons in the Prairie watershed of the Lower South Saskatchewan River basin (station 34), regardless of the accuracies of the precipitation data sets (station IX in Fig. 3a). In this case, the errors in the precipitation products played only a small in affecting the streamflow simulation.

The aim of the experimental design here was to reveal the quality of the precipitation products from a hydrological perspective while trying to isolate the effect of any compensations from parameter uncertainty. The basic assumptions were as follows: 1)

given the same model structure and process representation across the basin with one set of a priori parameter values, any differences in simulations shown in the above analysis would mainly come from precipitation errors, and 2) assuming the model structure and processes were represented correctly and the streamflow was measured with minimal uncertainty, any overestimations and underestimations of the precipitation amounts should symmetrically transfer to the errors in streamflow simulations. We observed that the model performance was not always in harmony with the errors assessed in the precipitation

products e.g., an overestimation of streamflow while precipitation amounts from different precipitation data sets were generally underestimated. Such cases, could imply that the errors from the precipitation products were outweighed by other errors. For instance, the results from the above streamflow-based precipitation comparison could have be affected by the choice of the a priori parameter set because the chosen parameter set might not have represented the correct values for the processes to function properly. Additionally, the baseflow representation in MESH with a conceptual bucket below the soil profile plays a major





role in how well the low flows are simulated, directly affecting how well the low-flow-season performance metrics perform. Consequently, errors from the process representation were introduced into the results. However, we acknowledged the difficulty of segregating the effects of errors from different sources, given the complexities of each sub-basin across the SaskRB and the insufficient understanding of all the hydrological processes and human activities in the basin.

### 6.3 Model calibration under the best performing precipitation data

In the previous section's analysis, CaPA was identified as the best-performing precipitation data set, and therefore, CaPA is used in this section for model calibration. As mentioned in Sect. 5.3, a sub-set of parameters from MESH horizontal, vertical and routing component were selected for model calibration for two reasons: (1) to reduce the number of model evaluations needed in calibration, thereby reducing the computational cost, and (2) to avoid compensation between error sources and to avoid improving streamflow simulation at the cost of degrading other model output. For example, we clearly presented that CaPA underestimates precipitation, and we wary of compensating for this underestimation by reducing the evapotranspiration amounts.

The parameter calibration results are presented in Fig. 5 using $F_{nse}$, $F_{lnse}$, and $F_{bias}$ performance metrics. Each plot in Fig. 5 has six components: (1&2) performance with a priori parameters during calibration and temporal validation periods (i.e., pre-cal and pre-val), (3&4) performance with calibrated parameters during calibration and validation periods (i.e., post-cal and post-val), (5&6) validation using independent stations (spatial validation) during calibration and validation periods (Fig. 5, these stations indicated in italics and blue colors).

In the Upper South Saskatchewan River basin, model performance improved after calibration in all of the cases; the calibrated model even performed better in the validation period than in the calibration period. A possible reason for this is that the basin becomes wetter in the validation period. When greater precipitation amounts drive the model, it can more easily match the streamflow. This can be demonstrated by the fact that the two "wet" precipitation products (CRU and GPCC) were able to produce better streamflow simulations (especially in summer) than "dry" CaPA without calibration (Fig. 4). The major improvement in model performance in the validation period is coming from spring, possibly because of reduced underestimation of CaPA and/or increasing temperatures that leading to more snowmelt and runoff.

A similar model performance was seen in the headwaters of the Bow River basin (stations 8-12), where the model produced better $F_{nse}$ during validation than in calibration. The improvement in model performance occurred in spring and summer in stations 10, 11, and12, and in autumn and winter in station 8. MESH performed consistently well in the lower part of the Bow River basin where major irrigation areas, diversions, and upstream reservoirs are found. This shows that our newly developed modules of irrigation, reservoirs, and flow diversion were capable of capturing the regulated streamflow in this section of the basin. The model also performed well in the validation stations (stations 14 and 15) during both calibration and validation, implying that the global calibrated parameter sets were able to capture the hydrological dynamics of the basin in both space and time.





A different model performance was observed in the Red Deer River basin. The model performed well in both calibration and validation stations during the calibration period. However, the model failed to simulate the streamflow during the validation period (stations 18-21), with $F_{nse}$ ranging from 0.13 to negative values and with a high positive $F_{bias}$ of 57 %. Furthermore, the model performance was worse after calibration in the validation period in which the model produced more streamflow than

that observed in summer and autumn (Fig. 5c). This worsening model performance could be due to two factors; (1) a possible failure in transferring the global calibrated parameter sets over space and time; (2) the introduction of errors during model calibration when the model failed to properly represent some of the hydrological processes (e.g., partitioning of underestimated precipitation amounts with low evapotranspiration, leading to high streamflow). A similar model performance was observed in the Eastern Saskatchewan River basin (stations 36 and 37), where the calibrated results in the validation period were worse

than uncalibrated ones. The reasons could be similar to those affecting the Red Deer River basin.

Although the model performed similarly poorly in the Battle River basin, the reasons for this were different. In this case, the underestimation of precipitation amounts played a major role, as discussed in the previous section. The model was able to reduce the negative $F_{bias}$ in the calibration stations after calibration but failed in the validation stations. Again, the hydrological dynamics of this basin were not well captured by the transfer of global calibrated parameter sets. However, the calibrated

streamflow simulations of these stations are generally of good quality and have good correlations with observations (Fig. S1 in supplementary materials).

Another model failure could be seen in the upper part of the Lower South Saskatchewan River basin (station 34) where the model performance was better without calibration in both the calibration and validation periods. Given the overestimation of precipitation amounts throughout the seasons (except summer), the model produced a high negative $F_{bias}$ for the individual

seasons after calibration. As with the other case, this overestimation could be attributed to the improper representation of some processes in this part of the basin.

The above evaluation strategy enabled us both to reveal the ability of the model to capture the hydrological response across the basin and to assess the global multiple-station calibration method in transferring the parameter sets in space and time. Overall, the model was able to perform well across the SaskRB with significant improvement in the median of $F_{nse}$ and $F_{lnse}$

in both calibration and validation periods, compared with those obtained pre-calibration. Similarly, the median of the $F_{bias}$ of all stations was reduced by more than 5 % in all cases. Additionally, the globally calibrated parameter set was able to provide reasonable streamflow simulations over validation stations and during the validation period. Despite some failures, global parameterization generally achieved better model performance across the basin in which around 60 % and 30 % of the stations resulted in $F_{nse}$ and $F_{lnse}$ greater than 0.5, respectively.

Despite these encouraging results, unsurprisingly, there were regional differences in model performance where the model failed to capture the hydrological regimes of some subbasins. Given the considerable dry bias in CaPA, it was not expected that the model could match the observed streamflow and result in a high negative $F_{bias}$. This was found to be especially true in sub-basins where precipitation errors played a dominant role (e.g., Battle). Poor model performance during the validation period in some sub-basins (e.g., Red Deer) might have occurred because the model was calibrated mainly to data measured





dry (drought) years (2003 – 2005) in the Prairies. Other precipitation and/or hydrological regimes were, therefore, not able to be captured by calibration.

## 6.4 Model validation on other fluxes and stores

This section describes further validation of the model performance conducted on other hydrologic fluxes and stores using additional observations of evapotranspiration (ET) and the total water storage (TWS) anomaly. Measurements of evapotranspiration are scarce and limited to flux tower observations. Modelled ET is often compared to flux tower observations with a very limited spatial coverage and/or to satellite-based gridded ET estimates. In this study, consistency in MESH simulated ET was assessed using NDVI-based ET estimates and point ET estimate in two-flux towers. The simulated TWS was compared against the GRACE satellite TWS observations. To reveal the level of performance differences, the validation of ET and TWS is presented for both before and after model calibration.

Figure 6 presents a box-and-whisker plot to compare daily observed and simulated ET for the period 2003-2009 at two flux tower sites: Old Jack Pine (OJP) and Old Black Spruce (OBS). Both sites are located in the downstream Boreal Plain (Fig. 1c). The flux-tower observations were compared to a needleleaf forest tile near the flux-tower sites. The majority of ET at both sites occurs from April to October, with a large portion in summer (June, July, and August). The median ET is close to zero for the low-temperature months from November to March. Fig. 6a shows that the pre and post-calibrated median and interquartile ranges of ET followed the observed seasonal ET pattern well. However, overestimation by MESH was observed from May to July, and the interquartile ranges were larger than those of the observed ranges. The median of post-calibrated ET was closer to observation than pre-calibrated ET.

Fig. 6b shows modelled and observed ET comparisons for OBS, with both pre- and post-calibration results, indicating comparable ET estimations against observations for median and interquartile ranges. Unlike OJP, ET underestimation was observed for the OBS site for July, August, and September. The difference between simulated and observed ET for the flux-tower sites is in part associated with scale mismatch between the MESH tile-scale simulations and flux point measurements (typically the coverage of a flux tower is at a scale of 100m), bias in meteorological forcing, and process representation of soil and vegetation in the model. Additionally, the observation from flux towers involves some adjustment for energy balance closure (Barr et al., 2012). Nevertheless, the overall performance at both flux towers showed similar performance quality as in a previous study by Davison et al. (2017) which conducted detailed MESH model calibration for a watershed where both observations are located. The results of our study indicates good performance of our globally optimized parameters values over the flux towers' region.

Regarding the gridded comparison of ET and TWS, a simple comparison was made by sampling the values of GRACE TWS and NDVI-based ET at the MESH grid scale using the nearest grid point approach. Figures 7a and 7b present gridded correlation coefficients between NDVI-based monthly ET and the MESH-simulated monthly ET before and after calibration, respectively. The comparison showed a reasonably good agreement between simulated ET and NDVI-based ET for both before and after calibration. Both had a high positive correlation (>0.8) over a large portion of the basin. The patterns of low





correlation were concentrated around the same regions in the basin, the majority of were in the lower portion of the Upper South Saskatchewan, with a small portion of the lower end of the Bow and upstream of the Lower South Saskatchewan sub-basins. The low correlation region is potentially related to irrigated areas (Fig. 1d) where more water is available from irrigation water diversion. However, other factors could also contribute to the difference between NDVI-based ET and MESH ET, such

as variation in the ET estimation methods and scale mismatch.

The gridded correlations between GRACE TWS and MESH-TWS (Figs 8a and 8b) showed a wide range of performance, but the general patterns of pre- and post-calibration performances are in agreement. A very good agreement was also observed between the basin averaged time series of the simulated and observed TWS anomaly (Fig. 8c). The calibrated MESH gridded TWS correlation results showed larger coverage (more grid cells) of high correlation than Pre-calibration TWS correlation

result. The high correlation values were observed around the central part of the SaskRB, including the Lower North and South Saskatchewan sub-basins as well as the lower portion of the Battle and Central North Saskatchewan subbasins. The headwater sub-basins (Upper South Saskatchewan, Bow, and Red Deer) showed high correlation around the upstream and downstream ends of the sub-basins. However, the gridded results revealed that the model did not perform well in some places, particularly in the Eastern part of the SaskRB, possibly because the process representation over open waters (lakes) is deficient in MESH,

as the inferior correlations are scattered around the water bodies of the SaskRB. Besides the MESH deficiency, it is notable that there are varying levels of uncertainty that come with GRACE data related to issues of scaling, filtering, and the removal of ocean, atmosphere and isostatic rebound signals (Seo, et al., 2006;; Landerer and Swenson, 2012).

**7 Conclusions and Implications**

In this work, we tested the capability of the MESH Hydrologic-Land Surface Model in capturing the hydrological dynamics

of the SaskRB using a multi-criterion, multi-station calibration approach, recognizing the input uncertainties from the climate forcing data and the complexities of the river system. We first compared different precipitation products and evaluated them against observed data to understand the error characteristics of the precipitation data sets (Sect. 6.1). Then, to minimize the effects of error compensation from model structural and parameter uncertainties and to reduce computational cost, we used the different precipitation datasets to drive MESH calibration with a default parametrization to evaluate their performance in

reproducing streamflow (Sect. 6.2). As seen in Sect. 6.2, we found that CaPA was the overall best-performing precipitation product, so we used it to calibrate MESH against observed streamflow. We evaluated the model by using four sets of criteria to test the transferability of the global calibrated parameter sets in space and time (Sect. 6.3). Finally, we took a further step to evaluate the model's ability in reproducing other water budget components by comparing them with additional information (Sect. 6.4). We conclude that:

1) Despite generally underestimating precipitation amounts across the SaskRB, CaPA showed the minimum precipitation errors in terms of magnitude and capturing the timing of precipitation.



2) Without calibration, the accuracy of the precipitation products was positively and magnitude of precipitation errors was negatively associated with the model performance in streamflow simulation in most of the sub-basins. The quality of precipitation products had a direct and immediate impact on the headwaters of the basin but the effects were dampened when going downstream. However, such associations did not hold in some sub-basins, reflecting the possibility that other errors (e.g., model structure and process representation) had potentially outweighed or offset the errors from the precipitation products.

3) In general, MESH was able to capture the hydrological response across the basin using the global multiple-station calibration method. Despite some failures, the global parameterization generally achieved better model performance across the basin in which around 60 % and 30 % of the stations resulted in $F_{nse}$ and $F_{lnse}$ greater than 0.5, respectively. Given the considerable dry bias in CaPA and the complexity of SaskRB, it is not surprising that there were regional differences in model performance, where the model failed to capture the hydrological regimes of some subbasins.

4) Further validation with complementary water budget component products (GRACE, NDVI-based ET, and two flux tower sites) showed good representation of ET for flux tower observation and good correlation with NDVI-based ET and GRACE observations. However, the validation revealed MESH's potential deficiency in capturing water storage over open water areas (i.e., in the Eastern Saskatchewan River basin).

The broader implications of this work include the followings:

The evaluation of precipitation uncertainties in the widely-used high-resolution datasets demonstrated that precipitation error is significant when compared to precipitation-gauge data, and different performance metrics showed a wide variation from one dataset to another. The datasets used and other similar products are important resources used extensively to identify regional to global freshwater resources and to assess the impact of future climate changes at different scales (Döll et al., 2003; Gosling and Arnell., 2011), However, as shown on this study domain, some of the data sets are considerably different from the ground-based observed values, raising concerns on the reliability of some of datasets for regional-scale applications, especially if they are selected without rigorous error examination against observations. Such significant precipitation uncertainty in some products can render H-LSMs and their parameterization ineffective. Therefore, our study emphasizes that a thorough evaluation of error characteristics of candidate precipitation datasets against gauge observation at a local or regional scale is a necessary step, before using the dataset for further application. Our analysis provides one of the possible ways to assess the precipitation data, yet, more work should be done to investigate the impacts of input uncertainties on H-LSMs, with more focus on other error characteristics such as the precipitation extremes.

Generally speaking, diagnosing model failures and potential error compensations among different model components in the presence of precipitation uncertainty is more challenging in case of conceptual global and catchment-scale hydrological models, in which different processes are approximated conceptually in a simplified way and parameter calibration is key to reliable streamflow simulation (Schmeid et al., 2014). On the other hand, diagnosing limitations of process-based H-LSMs faces less risk of error compensation and can be possibly done without calibration, because most H-LSMs parameters have a





direct physical interpretation, which introduces the possibility to define realistic default parametrization. This study showed that the approach of default parameterization could be used effectively to evaluate and identify suitable climate forcing and to reduce the risk of error compensation between different model components in the course of calibration. This reveals the potential benefits of process-based H-LSMs over conceptual hydrological models in the identification of the sources of errors

and future directions for model improvement.

Similar to hydrological models, calibration can help us achieve a better performance of H-LSMs, because of the inconsistency of scales between measurements and modeling grid, simplification of the surface heterogeneity representation and hydrological processes, parameter uncertainty in acceptable interval values, etc. For instance, Nasonova et al. (2009) demonstrated calibration significantly improved the H-LSM streamflow simulation for several Model Parameter Estimation Experiment

(MOPEX) catchments, and the performance of H-LSM on streamflow simulation after calibration was closely matched other hydrological models used in Duan et al. (2006). In a similar context, this study affirms that the utilization of effective calibration (parametrization) in an H-LSM makes it a robust modeling system, capable of producing improved streamflow simulation for large-scale basins with complex hydrological processes and highly managed environment. The H-LSM parameterizations enable the possibility of modeling ungauged watersheds through transferring optimal parameters, for

example based on similarity of landcover. However, care must be taken with the number of parameters to be calibrated, the usage of observation used in the calibration/validation processes, and the strategy of the calibration method.

Given the high-dimensionality of parameter spaces and the lack of confidence in defining the parameter values, one might argue that more (or even all) parameters should be calibrated to obtain better model performance. While model parameters are usually calibrated to streamflow only, calibrating more parameters could increase the risk of over-fitting and the chance of

introducing errors into other model stores and fluxes. Therefore, except the case of experimental watersheds with comprehensive observations, calibrating many parameters should not be encouraged as this essentially reduces the benefits of the 'process-based' nature of H-LSMs. With the increasing availability of observational information on hydrological components such as GRACE total water storage anomaly and satellite-based ET, one might utilize these data during data assimilation process or calibration alongside with observed streamflow. Uncertainties are often inherent within these data, but

rigorous assessments of the error distribution of these data are limited. This is especially true in the SaskRB (and in Canada generally), where ground observations are very sparse or simply non-existent. Thus, our study only utilized these data for temporal and spatial validation. Nonetheless, a more concerted effort is needed to assess and quantify the error characteristics of these data such that these data can be better utilized in H-LSM development and testing.

The demonstrated capability of a H-LSM in modelling a complex, highly-managed large-scale basin indicate its potential to

examine impacts of future climate and land-cover changes, and impacts on water resources management. Being an embedded component of climate and numerical weather prediction models is one of the assets for H-LSMs to seamlessly evaluate climate change impacts and perform flood forecasting and drought monitoring. Moreover, the modular organization of H-LSMs enhances the flexibility of incorporating new model process components. For example, the implementation of reservoir operation and estimation of irrigation demand can be readily updated when new information are available. This facilitates the



generation of different scenarios to evaluate the performance of existing reservoir operation under future changes and to assess the possibility of readjusting reservoir operation targets for adapting future changes. Similarly, the generation of future scenarios to examine the effects of land-cover changes such as the expansion of the irrigated area, glacier retreats, deforestation, and forest fires is possible by altering the gridded model setting. Yet, it is still a challenge to dynamically represent the land-

5 cover changes within the model. In addition, the ability of H-LSMs to output simulated variables regarding soil, vegetation, and snow at sub-grid scale (tile-based) provides finer details for better visualization over a region or a basin. As a result, H-LSMs are not only used as a modelling tool but also can be served as a platform to support policy making and water resources management.

**Acknowledgements**

10   This research was supported financially by the Canada Excellence Research Chair in Water Security and the Changing Cold Regions Network. The first author would like to acknowledge the financial support of the Saskatchewan Innovation Opportunity Scholarship. The authors would like to thank Daniel Princz for all the technical supports related to MESH model.





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





Figure 1. Map of study area, a) DEM, Sub-drainage, cities, and River network, b) Land-cover map along with Ecozone and Sub-drainage of SaskRB, c) Streamflow stations, Climate stations, and non-contributing map, d) Dams, irrigation districts, diversion, and irrigation abstraction points





w

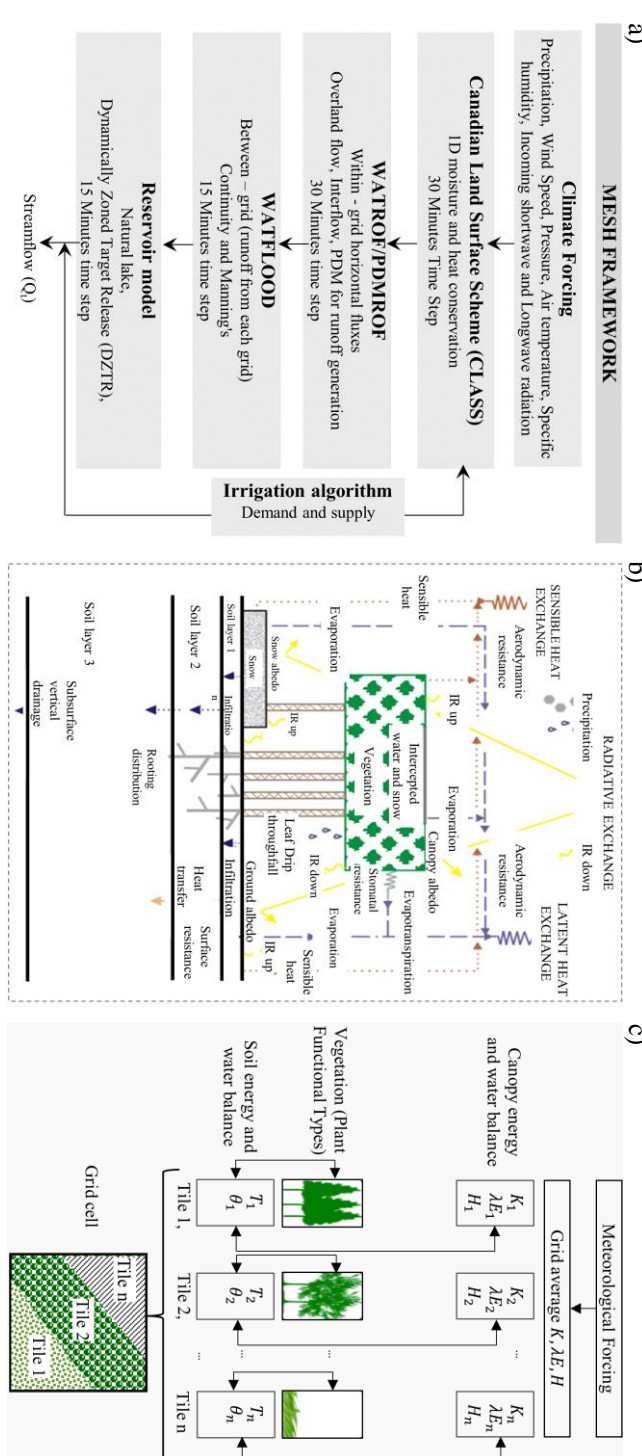

Figure 2. MESH model a) schematic diagram and b) CLASS schematic diagram, c) CLASS sub-grid structures for water and energy balance calculation

w



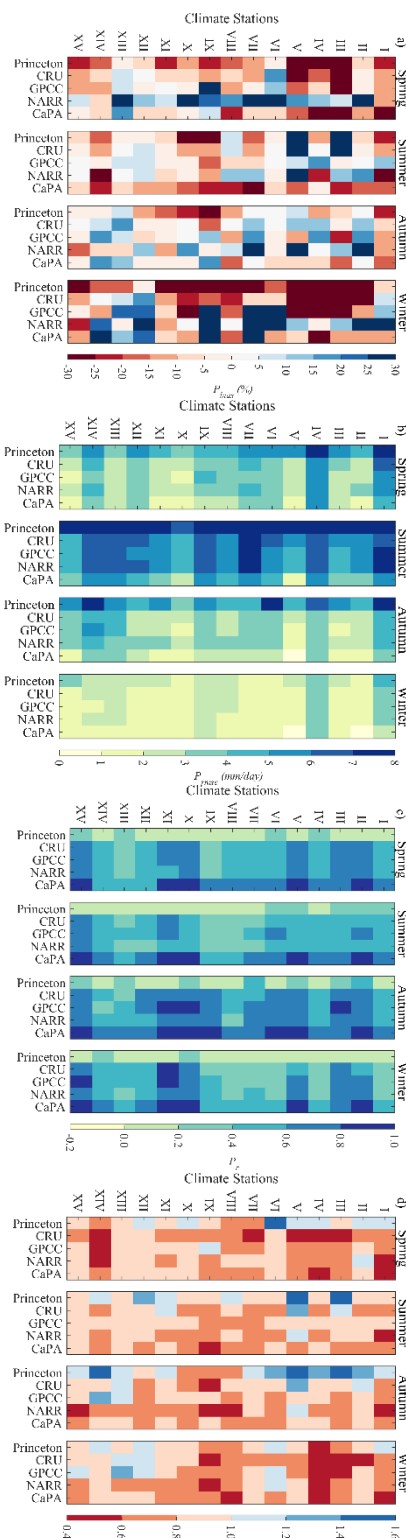

Figure 3. Metrics for precipitation of each type of gridded precipitation when evaluated against the precipitation-gauge station (station 1 to 15) in four seasons for the time period of 2002 to 2010. Each column indicates one gridded precipitation product and each row represents one climate station with a numerical code corresponding to climate station shown in Fig. 1c. a) The magnitude of the errors ($P_{rmse}$, b) strength and direction of the relationship between gridded products and precipitation-gauge stations ($P_r$), c) accuracy ($P_{bias}$), and d) amplitude of the variations ($P_{\sigma G/\sigma R}$)





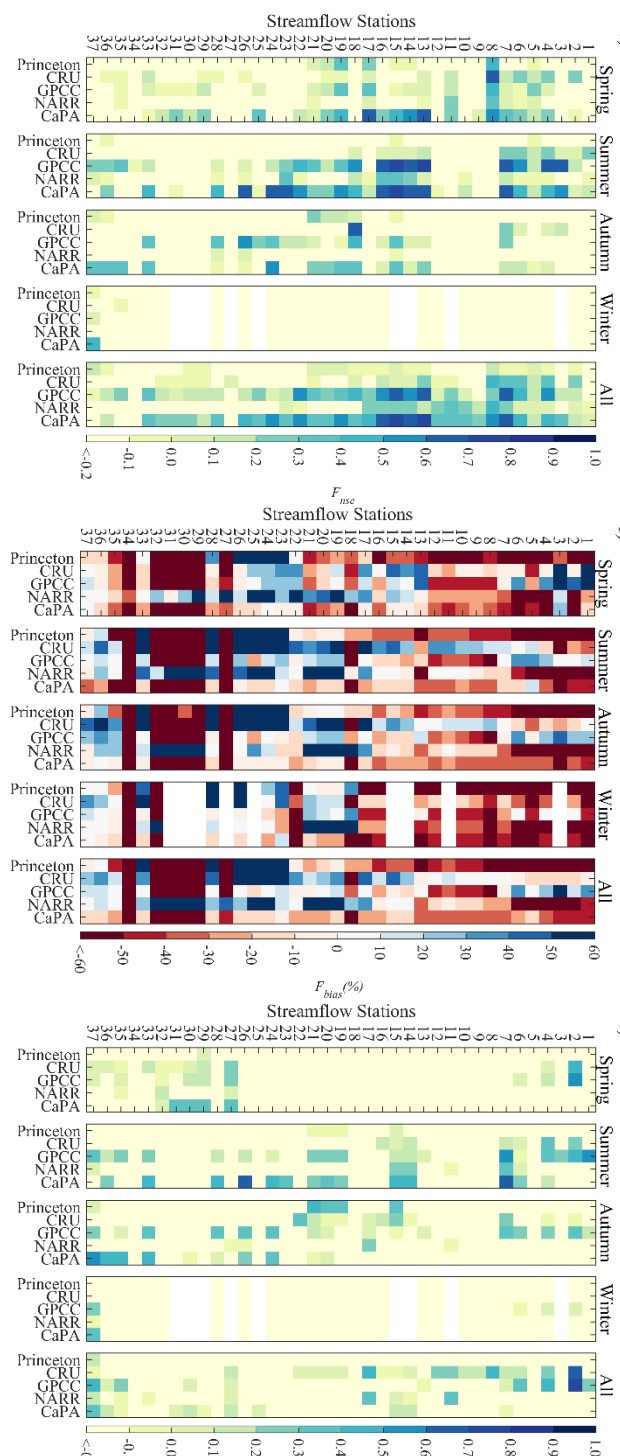

Figure 4. Streamflow simulation performance metrics for different precipitation data (10 years streamflow simulation against observations of multiple stations) a) $F_{nse}$, b) $F_{bias}$, and c) $F_{lnse}$. White indicates that no streamflow data are available as they have only seasonal observation. Stations 3, 11, 14, 15, 25, 27, 30, and 31 only have a seasonal observation with no observation during the winter period.





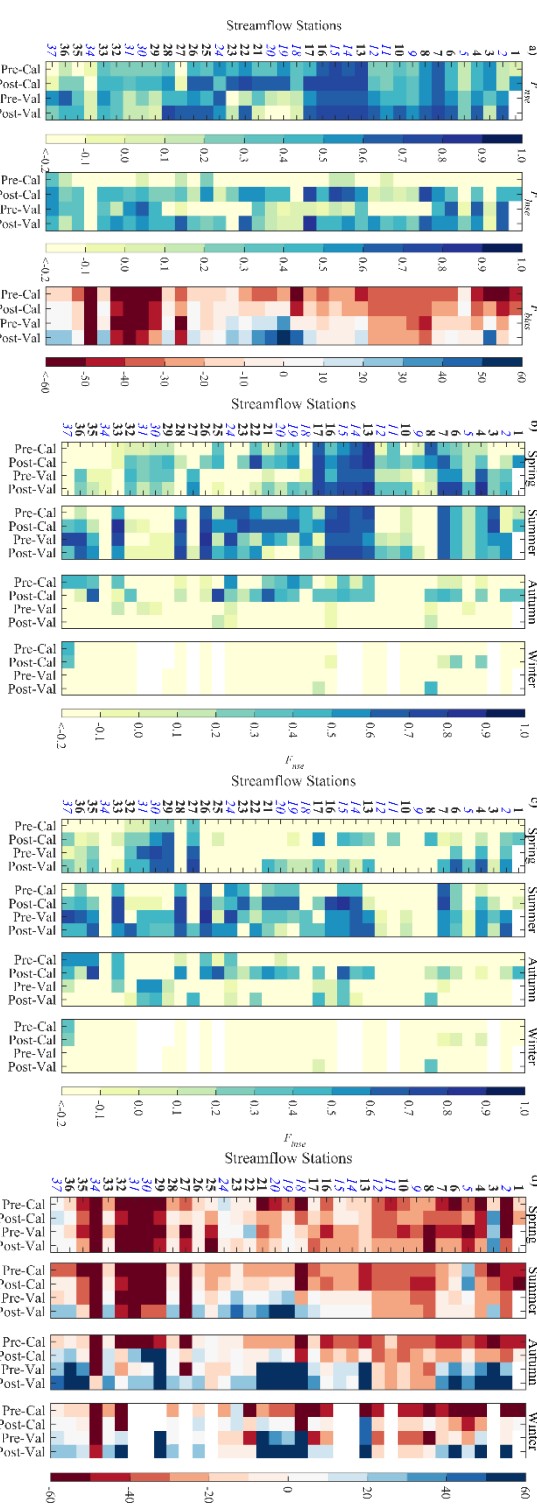

Figure 5. Streamflow simulation performance metrics before and after calibration. White indicates that no streamflow data are available as they have only seasonal observation. Calibration stations number has bold font weight and spatial validation stations numbers are italicized and blue color. Stations 3, 11, 14, 15, 25, 27, 30, and 31 only have a seasonal observation with no observation during the winter period.



Figure 6. Box plots of observed and simulated pre and post-calibrated daily ET in mm day⁻¹ for the period of 2003-2009 (a) Old Jack Pine site, (b) Old Black Spruce site

Figure 7. MESH evapotranspiration simulations correlation with NDVI-based evapotranspiration estimates a) pre-calibration b) post-calibration.


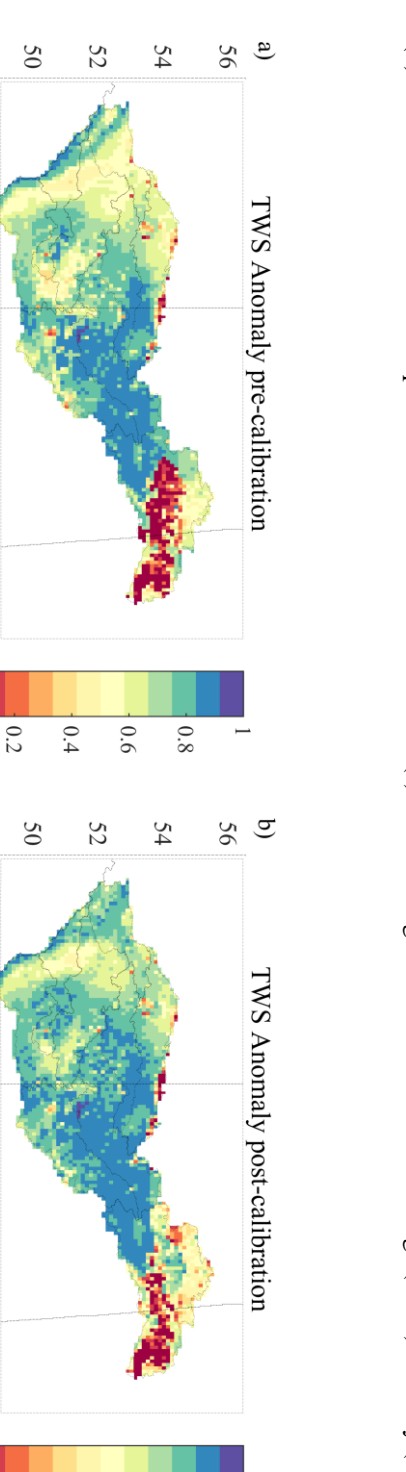

Figure 8. MESH model monthly total water storage (TWS) performance (a) correlation of MESH TWS pre-calibration with GRACE TWS (b) correlation of MESH TWS post-calibration with GRACE TWS (c) Basin average simulated total water storage (TWS) anomaly (cm)



Table 1: Precipitation products used for comparison

| Dataset | Full name | Type | Spatial resolution | Temporal resolution | Duration | Coverage | Reference |
|---|---|---|---|---|---|---|---|
| CaPA | Canadian Precipitation Analysis | Station-based multiple source | 300 arcsec (~0.0833°) /~10 km) | 6h | 2002-2014 | North America | Mahfouf et al. (2007) |
| Princeton | Global dataset at the Princeton University | Reanalysis-based multiple-source | 0.5°(~50 km) | 3h | 1901-2012 | Global | Sheffield et al. (2006) |
| WFDEI[CRU] | Water and Global change Forcing Data methodology applied to ERA-Interim [Climate Research Unit] | Reanalysis-based multiple-source | 0.5°(~50 km) | 3h | 1979-2012 | Global | Weedon et al. (2014) |
| WFDEI[GPCC] | Water and Global Change Forcing Data methodology applied to ERA-Interim [Global Precipitation Climatology Center] | Reanalysis-based multiple-source | 0.5°(~50 km) | 3h | 1979-2012 | Global | Weedon et al. (2014) |
| NARR | North American Regional Reanalysis | Reanalysis-based multiple-source | 32 km (0.3°) | 3h | 1979-2015 | North America | Mesinger et al. (2006) |



Table 2: Parameters and their corresponding ranges for calibration of the MESH model

| Parameter | Description | Range |
|---|---|---|
| *PDMROF parameters* | | |
| CMAX | Maximum storage parameter [m] | $(0.01, 5)^{C,IC,G}$ |
| B | Shape factor parameter [] | $(0.01, 10)^{C,IC,G}$ |
| *WATROF parameters* | | |
| MANN | Manning's roughness coefficient 'n' | $(0.05, 0.16)^{NF,BF,MF, SL}$ $(0.05, 0.16)^{G,GL,C,IC,B,BL}$ |
| KSAT | Saturated surface soil conductivity (m s$^{-1}$) | $(0.00001, 0.10)^{NF,BF,MF, SL}$ $(0.00001, 0.10)^{G,GL,C,IC,B,BL}$ |
| *River routing and baseflow parameters* | | |
| R2N | Channel Manning's rougness (N=9) | $(0.03, 0.16)$ |
| R1N | Overbank Manning's rougness (N=9) | $(0.03, 0.16)$ |
| LZF | Constant on lower zone function (N=9) | $(1.0E-06, 1.0E-04)$ |
| PWR | Exponent on the lower zone storage (N=9) | $(1.00, 3.00)$ |
| *CLASS parameters* | | |
| LAMAX | Annual maximum leaf area index | $(3.00, 10.00)^{BF}$ $(3.00, 5.00)^{CC, IC}$ $(3.00, 8.00)^{SL}$ $(3.00, 5.00)^{GG,GL}$ $(0.50, 3.00)^{NF}$ |
| LNZO | Natural logarithm of the roughness length | $(0.00, 1.10)^{BF}$ $(-2.53, -1.05)^{CC, IC}$ $(0.00, 1.10)^{SL}$ $(-3.91, -2.53)^{GG,GL}$ $(0.00, 0.69)^{NF}$ $(-4.60, -3.90)^{BB,BL}$ |
| ALVC | Average visible albedo when fully leafed | $(0.02, 0.10)^{BF}$ $(0.02, 0.10)^{CC, IC}$ $(0.02, 0.10)^{SL}$ $(0.02, 0.10)^{GG,GL}$ $(0.02, 0.10)^{NF}$ $(0.02, 0.10)^{BB,BL}$ |
| ALIC | Average near-infrared albedo when fully leafed | $(0.20, 0.40)^{BF}$ $(0.20, 0.40)^{CC, IC}$ $(0.20, 0.40)^{SL}$ $(0.20, 0.40)^{GG,GL}$ $(0.20, 0.40)^{NF}$ $(0.20, 0.40)^{BB,BL}$ |
| SDEP | Soil permeable depth (m) | $(0.7, 4.1)^{NF,BF,MF, SL,GL,BB,BL}$ |
| | | |
| Ranges for different land-cover types: NF= Needleleaf Forest, BF=Broadleaf Forest, MF= Mixed Forest, SL=Shrubland, G=Grassland, GL=Grassland lichen moss, B=Barrenland, BL=Barren lichen moss, C=Cropland, IC, Irrigated Cropland, N number of classification over the basin | | |



Table 3: Precipitation and streamflow performance metrics

| Precipitation performance metrics | | | |
|---|---|---|---|
| **Performance Measure** | **Symbol** | **Equation** | |
| Perccentage of Bias | $P_{bias}$ | $\dfrac{\sum_i^N (G_i - R_i)}{\sum_i^N (R_i)} \cdot 100$ | (1) |
| Root Mean Square Error | $P_{rmse}$ | $\sqrt{\dfrac{\sum_i^N (G_i - R_i)^2}{N}}$ | (2) |
| Correlation Coefficient | $P_r$ | $\dfrac{\sum_i^N (G_i - \bar{G})\,(R_i - \bar{R})}{\sqrt{\sum_i^N (G_i - \bar{G})^2}\,\sqrt{\sum_i^N (R_i - \bar{R})^2}}$ | (3) |
| Standard Deviation Ratio | $P_{\sigma_G/\sigma_R}$ | $\sqrt{\dfrac{\sum_i^N (G_i - \bar{G})^2}{N}} \Big/ \sqrt{\dfrac{\sum_i^N (R_i - \bar{R})^2}{N}}$ | (4) |

Note: $G$ and $R$ are the spatial average of the daily gridded precipitation product and the reference observation dataset (precipitation-gauge stations) respectively, $\bar{G}$ and $\bar{R}$ are the daily mean of gridded precipitation product and point station data over 2002–2010 respectively, i is the ith day of the season, and N is the total numbers of day in the season

| Streamflow performance metrics | | | |
|---|---|---|---|
| **Performance Measure** | **Symbol** | **Equation** | |
| Percentage of Bias flow | $F_{pbias}$ | $\dfrac{\sum (Q_{sim} - Q_{obs})}{\sum Q_{obs}} * 100$ | (5) |
| Nash Sutcliffe Efficiency on flow | $F_{nse}$ | $1 - \dfrac{\sum (Q_{obs} - Q_{sim})^2}{\sum (Q_{obs} - \overline{Q_{obs}})^2}$ | (6) |
| Nash Sutcliffe Efficiency on log flow | $F_{lnse}$ | $1 - \dfrac{\sum \big(ln(Q_{obs}) - ln(Q_{sim})\big)^2}{\sum \big(ln(Q_{obs}) - ln(\overline{Q_{obs}})\big)^2}$ | (7) |
| Correlation coefficients on TWS | $F_{rtws}$ | $\dfrac{\sum_i^N (TWS_{sim} - \overline{TWS_{sim}})\,(TWS_{obs} - \overline{TWS_{obs}})}{\sqrt{\sum_i^N (TWS_{sim} - \overline{TWS_{sim}})^2}\,\sqrt{\sum_i^N (TWS_{obs} - \overline{TWS_{obs}})^2}}$ | (8) |
| Correlation coefficients on ET | $F_{ret}$ | $\dfrac{\sum_i^N (ET_{sim} - \overline{ET_{sim}})\,(ET_{obs} - \overline{ET_{obs}})}{\sqrt{\sum_i^N (ET_{sim} - \overline{ET_{sim}})^2}\,\sqrt{\sum_i^N (ET_{obs} - \overline{ET_{obs}})^2}}$ | (9) |

$Q_{obs}$: Observed flow, $Q_{sim}$: Simulated flow, $\overline{Q_{obs}}$: Mean of observed flows

$TWS_{obs}$ and $TWS_{sim}$ are observed and simulated TWS; $\overline{TWS_{obs}}$ and $\overline{TWS_{sim}}$ are mean of observed simulated TWS; $ET_{obs}$ and $ET_{sim}$ are observed and simulated ET; $\overline{ET_{obs}}$ and $\overline{ET_{sim}}$ are mean of observed and simulated ET;


Table 4: Summary of a performance measure for precipitation products

| Performance Measure | Season | Precipitation Product | | | | |
|---|---|---|---|---|---|---|
| | | Princeton | WFDEI [CRU] | WFDEI [GPCC] | NARR | CaPA |
| $P_{bias}$ (%) | Spring | -17.90 | -8.36 | -2.96 | 19.11 | -11.37 |
| | Summer | -5.52 | 1.48 | 1.06 | -0.60 | -16.58 |
| | Autumn | -8.44 | 3.43 | 1.18 | 6.99 | -2.66 |
| | Winter | -31.68 | -15.86 | -0.68 | 24.71 | -0.33 |
| | Annual | -15.27 | -5.82 | -4.30 | 5.73 | -13.07 |
| $P_{rmse}$ (mm day$^{-1}$) | Spring | 5.22 | 3.52 | 3.41 | 3.54 | 2.78 |
| | Summer | 8.25 | 5.92 | 5.62 | 5.85 | 4.28 |
| | Autumn | 5.47 | 3.28 | 3.08 | 3.17 | 2.33 |
| | Winter | 2.68 | 1.96 | 1.97 | 1.91 | 1.54 |
| | Annual | 5.92 | 4.12 | 3.97 | 4.07 | 3.09 |
| $P_r$ (-) | Spring | 0.13 | 0.52 | 0.58 | 0.54 | 0.72 |
| | Summer | 0.17 | 0.47 | 0.53 | 0.47 | 0.72 |
| | Autumn | 0.18 | 0.62 | 0.66 | 0.60 | 0.80 |
| | Winter | 0.11 | 0.49 | 0.52 | 0.53 | 0.69 |
| | Annual | 0.17 | 0.51 | 0.56 | 0.52 | 0.73 |
| $P_{\sigma_G/\sigma_R}$ (-) | Spring | 0.99 | 0.71 | 0.81 | 0.79 | 0.78 |
| | Summer | 1.05 | 0.83 | 0.86 | 0.77 | 0.78 |
| | Autumn | 1.13 | 0.88 | 0.89 | 0.72 | 0.83 |
| | Winter | 0.87 | 0.75 | 0.85 | 0.76 | 0.70 |
| | Annual | 1.02 | 0.79 | 0.82 | 0.76 | 0.76 |