# Peer review of "Hydrologic-Land Surface Modelling of a Complex System under Precipitation Uncertainty: A Case Study of the Saskatchewan River Basin, Canada"

_Hydrology and Earth System Sciences, 2019_

## Referee Comment (RC1) · Anonymous Referee #1 · 25 Jul 2019

General Comments This paper describes the deployment of the MESH model, including assessment and selection of precipitation forcing, model calibration and model evaluation. The procedure is described using the Saskatchewan River Basin in western Canada as a case study. Although the description is thorough and clearly written, the work and methods described lack novelty. The results show that the chosen methodology simply works, not that it is an improvement (in terms of accuracy and/or efficiency) over some benchmark or baseline approach. Hence, the submitted work does not offer anything new to the science (theoretical or applied) of model development, calibra-

tion or validation. My overall evaluation is summarized as follows: 1. Does the paper address relevant scientific questions within the scope of HESS? No 2. Does the paper present novel concepts, ideas, tools, or data? No 3. Are substantial conclusions reached? No 4. Are the scientific methods and assumptions valid and clearly outlined? No 5. Are the results sufficient to support the interpretations and conclusions? No 6. Is the description of experiments and calculations sufficiently complete and precise to allow their reproduction by fellow scientists (traceability of results)? No 7. Do the authors give proper credit to related work and clearly indicate their own new/original contribution? Yes 8. Does the title clearly reflect the contents of the paper? No 9. Does the abstract provide a concise and complete summary? Yes 10. Is the overall presentation well structured and clear? Yes 11. Is the language fluent and precise? Yes 12. Are mathematical formulae, symbols, abbreviations, and units correctly defined and used? Yes 13. Should any parts of the paper (text, formulae, figures, tables) be clarified, reduced, combined, or eliminated? Yes 14. Are the number and quality of references appropriate? Yes 15. Is the amount and quality of supplementary material appropriate? Yes My recommendation is to reject the submitted work. Given the decision to reject the paper, in the following I will focus on major issues only.

Specific Comments The methodology for choosing the 'best' precipitation product is not defined and is, therefore, not reproducible. Although several evaluation metrics against ground-based observations (precipitation gauge and streamflow) are used, it is unclear how these results were combined and used to rank objectively the various products. As it currently stands, the choice of CaPA as superior to all other products is purely subjective.

The logic of ranking the accuracy of precipitation products by filtering them through and un-calibrated H-LSM to compare to streamflow observations is flawed. For this approach to be entirely valid, one must accept that errors in the precipitation data are propagated identically through the H-LSM for each product, independently of the chosen model parameters. Given that each precipitation product has different error

characteristics (bias, magnitude, amplitude variation, seasonality, etc.) it is plausible that the streamflow accuracy obtained via various precipitation product is, to some degree, conditioned upon the choice of 'default' parameter values (i.e. parameters may have default values, but in terms of the model performance, 'neutral' values may not exist). The authors may have chosen a default model parameter set that inadvertently optimised streamflow for the CaPA precipitation product.

The authors state that the rationale for choosing the best precipitation product is to derive the best (most accurate) calibrated H-LSM. However, the authors never end up demonstrating that this assumption is correct; this paper merely reinforces intuition, it does not reveal it as fact. Arguably, any number of precipitation products, once incorporated in the calibration process, could result in several parameterizations of MESH with very similar performance. In addition, by only using one precipitation product, the authors are not actually conducting what I would infer is "Hydrologic-Land Surface Modelling . . . under precipitation uncertainty . . .", as stated in the title.

In the first paragraph of the introduction it reads that the motivation behind the paper is predicated on the fact that the deployment and calibration requirements of Hydrologic-Land Surface Models (H-LSMs) differs markedly from Land Surface Models (LSMs), Hydrology Models (HMs) and Global Hydrology Models (GHMs). Without a clear definition/description of what constitutes/distinguishes these four type of models, treating H-LSMs as unique seems artificial. I am confident that decades of literature on the calibration of HMs does not need to be tossed because H-LSMs are so uniquely different (i.e. there is no need to start from scratch when discussing how to deploy an H-LSM).

Having as an objective the desire to 'improve the H-LSM parametrization using a state-of-the-art computationally efficient calibration approach . . ." seems quite trivial. There has been sufficient research with hydrologic modelling (whether that be an HM, GHM or LSM) to indicate that model calibration improves accuracy and is a necessary step. It is also arguable whether a calibration approach that relies on a single objective function constrained only by streamflow is actually state-of-the-art. In addition, the adopted

calibration approach does not actually tests the effectiveness of parameter transferability,as claimed. The use of independent streamflow gauges to evaluate the calibration does not test for transferability, as no parameters have been spatially transferred; they have in fact been calibrated in place (unless I missed something in the text). What is actually being tested using independent gauges is the spatial robustness of the calibrated model parameters.

———————————————

---

## Referee Comment (RC2) · Anonymous Referee #2 · 12 Aug 2019

Yassin and his colleagues reported their research work conducted by the MESH model. When I first read it, I was quickly saturated with too many trivial details, which are probably very important, but I couldn't remember any of them. When I read the paper again, I found several new things which were hidden among the ocean of numbers: 1) the authors improved the MESH model, by involving irrigation and flow diversion modules; 2) the new MESH model without calibration seems work well not only in streamflow simulation, but also to reproduce ET and TWS. Surprisingly, I did not find any reflection of these innovations in the title, also not any highlight in the main text.

More weird. I did not see the comparisons between the original MESH model and the new MESH model.

Generally, the authors did good research, and a lot of work. But the paper reads like an experimental report, rather than a research article. What can we learn from this paper? It is not necessary to show all the simulated data from the model. What messages did the authors want to deliver to readers? Did the authors want to report their finding something like: "CaPA is the best choice to conduct hydrological research in Saskatchewan River Basin"? If this is the main take-home message, I don't think this paper deserves to be accepted by HESS. Therefore, I suggest that a substantial major revision is needed before further consideration.

Other comments:

1. The paper is too long to read (40 pages), and quite easy to drain readers' energy and patience. It needs substantial shortening and condensing.

2. Figure 1 is not clear. Please make sure all the words in the figure can be read.

3. Many confusing points. For example, Page 16 line 31: "Such cases, could imply that the errors from the precipitation products were outweighed by other errors." If other errors outweigh precipitation uncertainty, is it convincing to use precipitation as input of the MESH to evaluate the quality of precipitation data?

---

## Author Comment (AC1) · 8 Nov 2019

**Responses to Reviewer 1 comments on Manuscript HESS-2019-207**

**Title:** Hydrologic-Land Surface Modelling of a Complex System under Precipitation Uncertainty: A Case Study of the Saskatchewan River Basin, Canada

**Authors:** Fuad Yassin et al

**Manuscript No:** hess-2019-207

The review comments are in regular bold typeface, while all responses are in italics and indented paragraphs.

**Response to Reviewer 1**

**General Comments:**
**This paper describes the deployment of the MESH model, including assessment and selection of precipitation forcing, model calibration and model evaluation. The procedure is described using the Saskatchewan River Basin in western Canada as a case study. Although the description is thorough and clearly written, the work and methods described lack novelty. The results show that the chosen methodology simply works, not that it is an improvement (in terms of accuracy and/or efficiency) over some benchmark or baseline approach. Hence, the submitted work does not offer anything new to the science (theoretical or applied) of model development, calibration or validation. My overall evaluation is summarized as follows: 1. Does the paper address relevant scientific questions within the scope of HESS? No 2. Does the paper present novel concepts, ideas, tools, or data? No 3. Are substantial conclusions reached? No 4. Are the scientific methods and assumptions valid and clearly outlined? No 5. Are the results sufficient to support the interpretations and conclusions? No 6. Is the description of experiments and calculations sufficiently complete and precise to allow their reproduction by fellow scientists (traceability of results)? No 7. Do the authors give proper credit to related work and clearly indicate their own new/original contribution? Yes 8. Does the title clearly reflect the contents of the paper? No 9. Does the abstract provide a concise and complete summary? Yes 10. Is the overall presentation well structured and clear? Yes 11. Is the language fluent and precise? Yes 12. Are mathematical formulae, symbols, abbreviations, and units correctly defined and used? Yes 13. Should any parts of the paper (text, formulae, figures, tables) be clarified, reduced, combined, or eliminated? Yes 14. Are the number and quality of references appropriate? Yes 15. Is the amount and quality of supplementary material appropriate? Yes My recommendation is to reject the submitted work. Given the decision to reject the paper, in the following I will focus on major issues only.**

> *We thank the reviewer for reviewing our manuscript and providing his/her valuable comments. Our understanding of the reviewer's reasons for rejecting our submitted work is that they are threefold: 1) our work does not address relevant scientific questions within the scope of HESS; 2) our work does not show results regarding improvements of model*

*development, calibration, validation over some benchmark approach, and 3) our work and methods lack novelty.*

*Regarding the first point, we wanted to strongly emphasize that our work was conducted based on the call of the special issue entitled 'Understanding and predicting Earth system and hydrological change in cold regions'. The special issue stated that "**the urgent need to understand the nature of the changes and to develop the improved modelling tools needed to manage uncertain futures… at multiple scales with a geographic focus on western Canada, including the Saskatchewan and Mackenzie River basins**". The objectives of our work are clearly in line with the call of this special issue, and the paper includes not only new insights into the modelling of a large scale river system in a cold climate, with limited data, but also advances in model capability. In particular, we demonstrated the advances in the diagnosis of an improved Canadian H-LSM (i.e. MESH with the inclusion of irrigation and flow diversion modules) in modelling the highly complex river system in western Canada with consideration of errors in precipitation data and their propagation through the model. In our view, and given the complexity and size of this basin, we presented an approach that is unique geographically and captures a very specialized application of a H-LSM that we have not seen in the literature.*

*With respect to the second point, we acknowledge that we did not show results regarding improvements of model development, calibration, validation over some benchmark approach. We understand that showing comparison results between improved model and original model is important to show the robustness and superiority of the improved model over the original one. However, it is vital to understand that there are no previous equivalent modeling developments for this unique system, and this contributes to the challenge in hydrological simulation of this region. In the introduction section [P4L25-34; P5L1-19], we demonstrated how our model development with MESH H-LSM is different from the available limited studies. Thus, our study, which includes extensive evaluation and diagnosis of the model deficiencies in a systematic and comprehensive way, should be considered a benchmarking attempt for a detail modeling of this complex basin, which has frankly been elusive or poorly represented in other studies. However, as pointed out by reviewer 2 we will include the comparison results between the improved model (including water management) and the original MESH (no water management) in the appendix.*

*For the last point, we highlight in the following the novelty and contribution of our work which includes a comprehensive three-stage evaluation strategy for an H-LSM.*

*Firstly, as we discussed in the Introduction, H-LSMs are rarely calibrated because of their large number of parameters the complex surface heterogeneities and complicated hydrologic and water management features of most river basins, which are heavily manged. In addition,he computational requirements escalate and multiply when considering the precipitation uncertainties (i.e. driving the H-LSM with multiple precipitation products). While calibration with multiple precipitation products could be possible with a more conceptual model (such as model that depends only on water balance and runs at coarser time resolution) without representing any water management modules*

*in a large-scale river basin (e.g. Eum et al., 2014), it is not pragmatic to conduct the same modelling exercise with a process-based H-LSM in a heavily-regulated river basin, such as the SaskRB. We tackled this challenge and offered new insights by presenting a thorough assessment of error characteristics of several candidate precipitation products using both direct and in-direct evaluation methods before calibration (first-stage evaluation). We consider this as novel aspect of our work.*

*Secondly, we note that H-LSM parameter estimation through calibration is still in its infancy. It is well known in the literature that "a priori" parameter values, typically based on classical approaches, are simply not an optimal solution at these scales. While arguably it might be sufficient to calibrate the model using only streamflow observations at basin outlets for smaller basins, it would be problematic to do so for large-scale basins because of the heterogeneities of the sub-basins across the whole basins (Faramarzi et al., 2016). We addressed this issue by presenting a multi-objective multi-station optimization approach using as many streamflow stations as possible (second-stage evaluation). We further evaluated the model performance by validating the spatial model outputs with additional information from the GRACE data set and two eddy-covariance field sites (third-stage evaluation). Constraining the H-LSM with multiple stations across a large-scale river basin and validating its spatial outputs have not been commonly done in previous studies, thus, we consider this as a further novel aspect of our work.*

*Also, we think that the length of the manuscript might affect the efficiency of delivering our main contribution to the readers (as pointed out by Reviewer 2). Therefore, we will vigorously shorten our manuscript and we will ensure we better highlight the significance and novelty of our work in the end of the Introduction Section, which is shown as follows:*

> *This study was* *conducted to address key questions raised in the special issue entitled "Understanding and Predicting Earth System and Hydrological Change in Cold Regions". The significance of the work is to demonstrate advances in diagnosis and calibration of an improved large-scale H_LSM (the*  *MESH model)*  *(with representation of water management)* *for the entire SaskRB including consideration of error propagation from the precipitation inputs by presenting a three-stage evaluation strategy*  *aimed to improve the understanding of the basin as a whole and create a test-bed for the simulation of alternative climate, land use and water management futures. Moreover, this work highlights that the current generation of land-surface models simply cannot capture the important hydrological controls in these complex systems.*

*Given the above response and the revision plan, we hope that Reviewer 1 could re-evaluate our manuscript and appreciate the novelty and contribution of our work.*

**Specific Comments:**

**(1) The methodology for choosing the 'best' precipitation product is not defined and is, therefore, not reproducible. Although several evaluation metrics against ground-based observations (precipitation gauge and streamflow) are used, it is unclear how these results were combined and used to rank objectively the various products. As it currently stands, the choice of CaPA as superior to all other products is purely subjective.**

*The reviewer's point is well-taken. We have assessed the precipitation products by direct and indirect evaluation methods without trying to combine or rank the results based on an overall performance measure. We believe that methodology for choosing the 'best' precipitation product is an ongoing research topic by itself and developing such an objective methodology is beyond the scope of this study, which is limited by the large computational requirements of using a physics-based model over such a large basin. For example model calibration for different precipitation products was not feasible, hence a pragmatic approach was taken, using default model parameters for the screening of precipitation products. Therefore, we chose CaPA for subsequent calibration based on the following judgement. While it is not easy to identify the overall best-performing precipitation data set using $P_{bias}$ or $P_{\sigma_G/\sigma_R}$, it is clearly seen that CaPA consistently outperformed other precipitation products at seasonal and annual scales when using $P_{rmse}$ and $P_r$ (see Table 4 in the manuscript). Additionally CaPA produced the overall highest seasonal and annual $F_{nse}$ across the SaskRB (see the following Table). Therefore, the choice of CaPA is not purely subjective.*

| Performance Measure | Season | Precipitation Product | | | | |
|---|---|---|---|---|---|---|
| | | Princeton | CRU | GPCC | NARR | CaPA |
| $F_{nse}$ (-) | Spring | 0.03 | 0.06 | 0.06 | 0.02 | 0.17 |
| | Summer | 0.00 | 0.04 | 0.24 | 0.04 | 0.28 |
| | Autumn | 0.02 | 0.03 | 0.10 | 0.00 | 0.12 |
| | Winter | 0.00 | 0.00 | 0.00 | 0.00 | 0.01 |
| | Annual | 0.01 | 0.06 | 0.25 | 0.08 | 0.35 |

*We will clarify the rationale of choosing CaPA for subsequent calibration by revising the first sentence of Section 6.3 [P17L6-7], as follows:*

*In the previous section's analysis, it is not easy to identify the overall best-performing precipitation data set when considering all performance metrics. Combining or ranking the results in a systematic and robust way could be possible, however, developing such methodology is out of the scope in this study.. We chose the best-performing precipitation data set based on superior performance on multiple performance measures. It is clearly seen that CaPA consistently outperformed other precipitation products at seasonal and annual scales when using $P_{rmse}$ and $P_r$ (Table 4). A similar situation is seen when CaPA produced the overall highest seasonal and annual $F_{nse}$ across the SaskRB (Fig. 4). Therefore, CaPA is used in this section for model calibration.*

**(2) The logic of ranking the accuracy of precipitation products by filtering them through and un-calibrated H-LSM to compare to streamflow observations is flawed. For this approach to be entirely valid, one must accept that errors in the precipitation data are propagated identically through the H-LSM for each product, independently of the chosen model parameters. Given that each precipitation product has different error characteristics (bias, magnitude, amplitude variation, seasonality, etc.) it is plausible that the streamflow accuracy obtained via various precipitation product is, to some degree, conditioned upon the choice of 'default' parameter values (i.e. parameters may have default values, but in terms of the model performance, 'neutral' values may not exist). The authors may have chosen a default model parameter set that inadvertently optimised streamflow for the CaPA precipitation product.**

*We appreciate the value of the reviewer's comment on the choice of default parameter values. The values of the default parameter set were chosen from direct physical interpretation of those parameters (i.e. according to the CLASS manual in our case), if not, from the available literature for the model. The decision to select the parameter values was made purely on how realistic the values represent the underlying processes, regardless of the precipitation inputs. Therefore, by assuming the processes are functioned properly, the impact of default parameter values on streamflow performance by different precipitation products should be minimum. However, we acknowledge the point of the reviewer in which the use of default parameters has possible limitation and might affect the selection of the best product in subtle way. We will acknowledge the limitation of choosing the default parameter values on streamflow performance by extending the discussion in Section 6.2*

**(3) The authors state that the rationale for choosing the best precipitation product is to derive the best (most accurate) calibrated H-LSM. However, the authors never end up demonstrating that this assumption is correct; this paper merely reinforces intuition, it does not reveal it as fact. Arguably, any number of precipitation products, once incorporated in the calibration process, could result in several parameterizations of MESH with very similar performance. In addition, by only using one precipitation product, the authors are not actually conducting what I would infer is "Hydrologic-Land Surface Modelling . . . under precipitation uncertainty . . .", as stated in the title.**

*We appreciate the value of the reviewer's comment on the validity of the assumption we made in our work and the concern for model equifinality. First of all, we wanted to reiterate the fact that calibrating a process-based H-LSM for a large-scale heavily-managed river basin is very computational intensive (P11L4-10 in the manuscript). It is possible but not pragmatic to do so when accounting for the precipitation uncertainties. Secondly, calibrating the model with other precipitation products might have similar performance to the best performing precipitation product. However, such good performance would likely be a result of error compensation during calibration and, more importantly, not give the right answers for the right reasons. Therefore, we presented the first-stage evaluation as a screening stage to illuminate the error characteristics of the precipitation products and*

*help eliminate possible 'bad' products before calibration. Regarding the title, we are considering revising the title, perhaps to something like "Hydrologic-Land Surface Modelling of Complex, Heavily Managed Watershed Systems: Addressing Human Interventions and Precipitation Error". We welcome the reviewer's opinions and suggestions on the title.*

**(4) In the first paragraph of the introduction it reads that the motivation behind the paper is predicated on the fact that the deployment and calibration requirements of Hydrologic Land Surface Models (H-LSMs) differs markedly from Land Surface Models (LSMs), Hydrology Models (HMs) and Global Hydrology Models (GHMs). Without a clear definition/description of what constitutes/distinguishes these four type of models, treating H-LSMs as unique seems artificial. I am confident that decades of literature on the calibration of HMs does not need to be tossed because H-LSMs are so uniquely different (i.e. there is no need to start from scratch when discussing how to deploy an H-LSM).**

*We understand the concern of the reviewer and we think that mentioning different types of models in the Introduction section might create confusion and make the motivation of the paper unclear. Accordingly, we will revise the first paragraph in the Introduction section [P2L1-10], which is shown as follows:*

*During the past few decades, the development of hydrological models (HMs) for large-scale application (~$10^3$-$10^6$ km$^2$) has  expanded in scope and complexity because of emerging water security challenges (Eagleson 1986; Clark et al., 2015; Döll et al., 2003; Vörösmarty et al., 2000). There are many large-scale HMs, which broadly fall into two categories, namely, Global Hydrological Models (GHMs) and Land Surface Models (LSMs) (Döll et al., 2016; Gudmundsson et al., 2012). GHMs are based on conceptual model approaches dominantly derived from catchment-scale HMs and aim to improve scale-appropriate process representations mainly for water management purposes. LSMs, on the other hand, are originally built to provide lower boundary conditions to climate models without considering any dominant hydrological processes, such as horizontal hydrological fluxes, subsurface lateral water movement, and river flow routing. To improve the utility of LSMs for large-scale hydrological modelling purposes, LSMs have taken the advantages of GHMs and HMs and become more sophisticated. They  have increasingly integrated dominant hydrological processes and, more recently,  (Archfield et al., 2015; Davison et al., 2016).  have included irrigation and water management modules (Haddeland et al., 2006;*

*Voisin et al. 2013a, 2013b; Pokhrel et al., 2017)  . The integration of these various processes has enabled LSMs to be used in support of a wide range of hydrological applications, in which they are referred to as Hydrologic-Land Surface Models (H-LSMs) (Pietroniro et al., 2007). Although H-LSMs have made steady advances in representing hydrologic processes and incorporating human impacts on the terrestrial water cycle, the investigation of input data uncertainty and parameter estimation through calibration for large-scale basins has been limited and is not common practice with H-LSM models compared to their extensive use by the catchment hydrological modeling community.*

**(5)** **Having as an objective the desire to 'improve the H-LSM parametrization using a state of-the-art computationally efficient calibration approach . . ." seems quite trivial. There has been sufficient research with hydrologic modelling (whether that be an HM, GHM or LSM) to indicate that model calibration improves accuracy and is a necessary step. It is also arguable whether a calibration approach that relies on a single objective function constrained only by streamflow is actually state-of-the-art. In addition, the adopted calibration approach does not actually tests the effectiveness of parameter transferability, as claimed. The use of independent streamflow gauges to evaluate the calibration does not test for transferability, as no parameters have been spatially transferred; they have in fact been calibrated in place (unless I missed something in the text). What is actually being tested using independent gauges is the spatial robustness of the calibrated model parameters.**

*We thank the reviewer for raising his/her concerns on the objective of our work and our calibration approach. Regarding the objective, we acknowledge that the current presentation of the objectives was not fully reflected the main goal of our study. After considering both reviewers' comments, we will revise the presentation of our study objectives [P4L1-17] to better reflect our work, which is shown as follows:*

*The aim of this paper is to present a three-stage evaluation strategy for   a physically-based H-LSM  over a highly-managed, large-scale basin, using state-of-the-art calibration strategies and multiple data sources to enable quantification of modelling uncertainty. Such analysis is essential to benchmark model performance, to examine water security vulnerabilities under future conditions, to serve as a test-bed (experimental basin) for the improvement testing of different model process, and to evaluate new datasets. Additionally, such analysis helps to inform H-LSM applications for hydrologic operational forecasts and the management of large-scale basin water resources.*

*The three-stage evaluation strategy consists of three specific objectives, as follows* *:*

- *To identify a suitable precipitation dataset for the H-LSM modeling based on: (1) precipitation error characteristics against ground-based observation, and (2) performance measure criteria based on streamflow simulation when used to drive default parametrized H-LSM.*
-
-
- *To conduct a multi-objective multi-station optimization approach,*  *and evaluate the effectiveness of parameter transferability through validation in time and space, using independent multiple streamflow gauges not used in calibration.*
- *To test the model performance using multiple sources of observational information on model storage and output fluxes, to ensure that the optimal parameters obtained are as realistic as possible (giving the "right answers for the right reasons") without error compensation across multiple outputs.*

*Please note that it is possible in principle to expand the optimization to a fully multi-objective problem, but this requires considerably many more optimization evaluations to identify the optimal Pareto solutions than finding a single solution for the aggregated optimization criterion. Due to computational challenges, running MESH with multi-objective optimization is not desirable for large-scale basin like SaskRB. We highlighted the computational challenge of running MESH on Page 11 [lines 6-10 and 21-27]. Thus, to reduce the computational burden and achieve effective parametrization, it is more desirable to deploy approaches like DDS.*

*Regarding parameter transferability, we want to emphasize that the parameters have been spatially transferred in the validation process in two ways. The first way is based on physical similarity of the basins (Patil and Stieglitz, 2015). Because of the sub-grid heterogeneity representation in MESH, the parameters, which are similar to many other LSMs, are tied to the vegetation types. For instance, parmeters for needleleaf forest at the calibrated stations would be transferred to basins where needleaf forest exists in the validation process. The second way is to transfer the parameters across different spatial scales (i.e. transferring parameters calibrated at the outlets to sub-basins within the basin, or vice versa) (Wöhling et al. 2013). We followed the approach used in the study of Wöhling et al. (2013) which could be considered as an internistical validation of the reliability of simulation of streamflow at any points of the basin.*

*In addition, our gridded spatioal validation evaluation on TWS and ET support the parameter transfer evaluation based on the first case that is parameter transfer based on physical similarity. Furthermore, it is important to note that our case study area is highly heterogeneous and large (around 400,000km$^2$ which is roughly the size of Germany), and contains water management complexity. We believe showing good simulation at this scale in both validation evaluation approach can be considered as proper evaluation of model transferability. We will clarify the use of the term "parameter transferribilty" in the revised manuscript. However, we would be open to the suggestions and decisions by the Editor if the clarification is still causing confusion.*

*References*

*Archfield SA, Clark M, Arheimer B, Hay LE, McMillan H, Kiang JE, Seibert J, Hakala K, Bock A, Wagener T, et al. 2015. Accelerating advances in continental domain hydrologic modeling. Water Resources Research **51** (12): 10078–10091 DOI: 10.1002/2015WR017498*

*Davison B, Pietroniro A, Fortin V, Leconte R, Mamo M, Yau MK, Davison B, Pietroniro A, Fortin V, Leconte R, et al. 2016. What is missing from the prescription of hydrology for land surface schemes? Journal of Hydrometeorology **17** (7): 2013–2039 DOI: 10.1175/JHM-D-15-0172.1*

*Döll P, Kaspar F, Lehner B. 2003. A global hydrological model for deriving water availability indicators: model tuning and validation. Journal of Hydrology **270** (1–2): 105–134 DOI: 10.1016/S0022-1694(02)00283-4*

*Döll P, Douville H, Güntner A, Müller Schmied H, Wada Y. 2016. Modelling Freshwater Resources at the Global Scale: Challenges and Prospects. Surveys in Geophysics **37** (2): 195–221 DOI: 10.1007/s10712-015-9343-1*

*Eagleson PS. 1986. The emergence of global-scale hydrology. Water Resources Research **22** (9S): 6S-14S DOI: 10.1029/WR022i09Sp0006S*

*Haddeland I, Skaugen T, Lettenmaier DP. 2006. Anthropogenic impacts on continental surface water fluxes. Geophysical Research Letters **33** (8): L08406 DOI: 10.1029/2006GL026047*

*Patil SD, Stieglitz M. 2015. Comparing spatial and temporal transferability of hydrological model parameters. Journal of Hydrology **525**: 409–417 DOI: 10.1016/J.JHYDROL.2015.04.003*

*Pokhrel YN, Felfelani F, Shin S, Yamada TJ, Satoh Y. 2017. Modeling large-scale human alteration of land surface hydrology and climate. Geoscience Letters **4** (1): 10 DOI: 10.1186/s40562-017-0076-5*

*Vörösmarty CJ, Green P, Salisbury J, Lammers RB, Falkenmark M. 2000. Global Water Resources: Vulnerability from Climate Change and Population Growth. Science **289** (5477): 284–288 DOI: 10.1126/science.289.5477.284*

*Voisin N, Li H, Ward D, Huang M, Wigmosta M, Leung LR. 2013a. On an improved sub-regional water resources management representation for integration into earth system models. Hydrology and Earth System Sciences* **17** *(9): 3605–3622 DOI: 10.5194/hess-17-3605-2013*

*Voisin N, Liu L, Hejazi M, Tesfa T, Li H, Huang M, Liu Y, Leung LR. 2013b. One-way coupling of an integrated assessment model and a water resources model: evaluation and implications of future changes over the US Midwest. Hydrology and Earth System Sciences* **17** *(11): 4555–4575 DOI: 10.5194/hess-17-4555-2013*

*Wöhling T, Samaniego L, Kumar R. 2013. Evaluating multiple performance criteria to calibrate the distributed hydrological model of the upper Neckar catchment. Environmental Earth Sciences* **69** *(2): 453–468 DOI: 10.1007/s12665-013-2306-2*

---

## Author Comment (AC2) · 8 Nov 2019

**Responses to Reviewer 2 comments on Manuscript HESS-2019-207**

**Title:** Hydrologic-Land Surface Modelling of a Complex System under Precipitation Uncertainty: A Case Study of the Saskatchewan River Basin, Canada

**Authors:** Fuad Yassin et al

**Manuscript No:** hess-2019-207

The review comments are in regular bold typeface, while all responses are in italics and indented paragraphs.

**Response to Reviewer 2**

**General Comments:**

**Yassin and his colleagues reported their research work conducted by the MESH model. When I first read it, I was quickly saturated with too many trivial details, which are probably very important, but I couldn't remember any of them. When I read the paper again, I found several new things which were hidden among the ocean of numbers: 1) the authors improved the MESH model, by involving irrigation and flow diversion modules; 2) the new MESH model without calibration seems work well not only in streamflow simulation, but also to reproduce ET and TWS. Surprisingly, I did not find any reflection of these innovations in the title, also not any highlight in the main text. More weird. I did not see the comparisons between the original MESH model and the new MESH model.**

> *We thank the reviewer for reviewing our manuscript and providing his/her valuable comments. In particular, we appreciate that the reviewer acknowledges there are certain novelties and innovations in this work. We agree that the presentation of the materials and the highlights was sub-optimal in the original version, and there were probably too many details presented that were not warranted. Therefore, we intend to significantly shorten the manuscript in a possible revised version and focus mainly on the innovations. Further, we understand that showing comparison results between the improved model and the original model is important to show the robustness and superiority of the improved model over the original. In response to the reviewer's comments (also pointed out by Reviewer 1), we will include the comparison results between the improved model (including water management) and the original MESH (no water management) in the appendix and keep our main focus on the comprehensive three-stage evaluation strategy for the improved MESH model in the main text.*

*Regarding the title, we are considering revising the title, perhaps to something like "Hydrologic-Land Surface Modelling of Complex, Heavily Managed Watershed Systems: Addressing Human Interventions and Precipitation Error". We welcome the reviewer's opinions and suggestions on the title.*

**Generally, the authors did good research, and a lot of work. But the paper reads like an experimental report, rather than a research article. What can we learn from this paper? It is not necessary to show all the simulated data from the model. What messages did the authors want to deliver to readers? Did the authors want to report their finding something like: "CaPA is the best choice to conduct hydrological research in Saskatchewan River Basin"? If this is the main take-home message, I don't think this paper deserves to be accepted by HESS. Therefore, I suggest that a substantial major revision is needed before further consideration.**

*We appreciate the value of the reviewer's comment on our work and we agree that the length of the manuscript might affect the efficiency of delivering our main contribution to the readers. Here, we highlight the significance of our work the advances it offers in the the diagnosis of an improved Canadian H-LSM (i.e. MESH with the inclusion of irrigation and flow diversion modules) in modelling the highly complex river system in western Canada (i.e. SaskRB) with the consideration of errors propagation from the precipitation inputs. These advances were shown by presenting a three-stage evaluation strategy for an improved H-LSM. The first-stage evaluation was to assess the error characteristics of several precipitation candidates through the direct and in-direct evaluation methods before calibration. Such evaluation is rarely done in previous studies, especially for process-based H-LSMs that model large-scale heavily-regulated basins. The second-stage evaluation was to conduct a multi-objective multi-station optimization approach using as many streamflow stations as possible for improved model performance and the third-stage evaluation was to further evaluate the model performance by validating the spatial model outputs with additional information from the GRACE data and two evapotranspiration data. Calibrating an H-LSM with multiple stations across a large-scale river basin and validating its spatial outputs are not commonly done in previous studies mainly because H-LSM parameter estimation through calibration is still in its infancy stage. In response to the reviewer's comments, we will vigorously shorten our manuscript (as shown in responding comment 1 in Other Comments Section) and we will highlight the significance of our work in the end of the Introduction Section [P5L17-19].*

*In addition, inspired by the Reviewer 1's comment 5, we think that the current presentation of our objectives might not fully reflect the main goal of our study and hence reduce the creditability of delivering the main messages to the readers. We will revise the presentation of our study objectives [P4L1-17] to better reflect our work, which is shown as follows:*

*The aim of this paper is to present a three-stage evaluation strategy for  a physically-based H-LSM  over a highly-managed, large-scale basin, using state-of-the-art calibration strategies and multiple data sources to enable quantification of modelling uncertainty. Such analysis is essential to benchmark model performance, to examine water security vulnerabilities under future conditions, to serve as a test-bed (experimental basin) for the improvement testing of different model process, and to evaluate new datasets. Additionally, such analysis helps to inform H-LSM applications for hydrologic operational forecasts and the management of large-scale basin water resources.*

*The three-stage evaluation strategy consists of three specific objectives, which is shown as follows :*

- *To identify a suitable precipitation dataset for the H-LSM modeling based on: (1) precipitation error characteristics against ground-based observation, and (2) performance measure criteria based on streamflow simulation when used to drive default parametrized H-LSM.*
-
-
- *To conduct a multi-objective multi-station optimization approach,  and evaluate the effectiveness of parameter transferability through validation in time and space, using independent multiple streamflow gauges not used in calibration.*
- *To test the model performance using multiple sources of observational information on model storage and output fluxes, to ensure that the optimal parameters obtained are as realistic as possible (giving the "right answers for the right reasons") without error compensation across multiple outputs.*

*Subsequently, we will also revise the abstract [P1L14-26] to reflect the changes that address both reviewers' comments:*

> *A three-stage evaluation strategy  of the MESH model performance was carried out . First, the reliability of multiple precipitation products was evaluated against climate station observations and based on their performance in simulating streamflow across the basin when forcing the MESH model with a default parameterization. Second, a state-of-the-art multi-criteria multi-station optimization  approach was*

*applied, using* multiple streamflow gauge stations across the river basin  *for calibration* . *Third, various observational information including storage and fluxes were used for further model validation.* ~~The first analysis shows that the quality of precipitation products had a direct and immediate impact on simulation performance for the basin headwaters but effects were dampened when going downstream. In particular, the Canadian Precipitation Analysis (CaPA) performed the best among the precipitation products in capturing timings and minimizing the magnitude of error against observation, despite a general underestimation of precipitation amount. The subsequent analyses show that the MESH model was able to capture observed responses of multiple fluxes and storage across the basin using a global multi-station calibration method. Despite poorer performance in some basins, the global parameterization generally achieved better model performance than a default model parameterization. Validation using storage anomaly and evapotranspiration generally showed strong correlation with observations, but revealed potential deficiencies in the simulation of storage anomaly over open water areas.~~ *The first-stage evaluation revealed the different error characteristics of precipitation datasets that are directly propagate to H-LSM modeling, and allowed identify the better precipitation dataset candidates for better H-LSM modelling. The comprehensive analysis in the subsequent stages demonstrated the capability of MESH (H-LSM) to model highly regulated and complex basin as well the possibility to improve the model simulation through global multi-station parametrization than a default model parametrization, while revealing potential deficiencies in simulation of water storage anomaly over open water areas.*

**Other Comments:**

(1) **The paper is too long to read (40 pages), and quite easy to drain readers' energy and patience. It needs substantial shortening and condensing.**

*We agree that the manuscript is lengthy and we will vigorously shorten our manuscript to improve the readability. We will focus on shortening and revising the Results and Discussion Section.*

(2) **Figure 1 is not clear. Please make sure all the words in the figure can be read.**

*Thanks for the reviewer's comment. We will revise Figure 1 to make sure all the words are visible to the readers.*

(3) **Many confusing points. For example, Page 16 line 31: "Such cases, could imply that the errors from the precipitation products were outweighed by other errors." If other errors outweigh precipitation uncertainty, is it convincing to use precipitation as input of the MESH to evaluate the quality of precipitation data?**

*We appreciate the value of the reviewer's comment on the validity of evaluating the quality of precipitation data through a process-based H-LSM (as also commented by Reviewer 1). We wanted to highlight the fact that the rationale behind this statement is based on two assumptions we made (as stated in the manuscript P16L24-29) and the belief that we are more confident in a process-based H-LSM in which we have explicitly represented the known hydrological processes and water management practices in the model using our best knowledge and understanding on those processes (e.g. snow processes, vegetation and soil processes, irrigation). We think that the basis of the two assumptions is reasonable, and therefore, we could obtain new information and insight about the quality of the precipitation products through a process-based H-LSM. However, we are also very aware of the deficiencies of the MESH model through our work (as discussed in Sections 6.3 and 6.4) and we acknowledge that caution is needed when interpreting the results of precipitation products, particularly in any sub-basins where model deficiencies are found.*

*We will clarify the validity of evaluating the quality of precipitation data through a process-based H-LSM by extending the discussion in Section 6.2 [P1623-34; P17L1-4].*

---

## Editor Comment (EC1) · Bettina Schaefli (Editor) · 18 Nov 2019

Both reviewers were very critical about the form and the content of this manuscript, one recommended rejection, one major revisions. Besides the structure and the length of the paper, both reviewers critically assessed the lack of novelty and the validity of methods used in the paper.

With this respect, the answers of the authors in the public discussion are at times very limited, and not always sufficient to answer the significant concerns of the reviewers.

[Figure]

An example is the concern expressed by reviewer 1 on the lack of evidence for having chosen the best precipitation product with the approach presented in the paper (comment 3). The answer to this critical comment simple states: "(..)we wanted to reiterate the fact that calibrating a process-based H-LSM for a large-scale heavily-managed river basin is very computational intensive. It is possible but not pragmatic to do so when accounting for the precipitation uncertainties. Secondly, calibrating the model with other precipitation products might have similar performance to the best performing precipitation product. However, such good performance would likely be a result of error compensation during calibration and, more importantly, not give the right answers for the right reasons." Rather than addressing the actual reviewer concern, the authors suggest here to change the title of the manuscript.

While in my view, the answers to the reviewers concerns are not sufficient at times, I agree with the authors that the content of the paper goes beyond a simple case study and certainly contains novel aspects in terms of modelling of a large complex catchment, especially in cold environments.

However, adapting the manuscript and convincingly explaining and demonstrating the developed modelling strategy will certainly result in a deep reorganisation before publication in HESS. Considering in addition, that the reviewers are not willing to re-review the revised version, I do not recommend the submission of a revised manuscript but recommend the submission of a new manuscript, to be handled by a new Editor.